# Mapping causal links between prefrontal cortical regions and intra-individual behavioral variability

Farshad Alizadeh Mansouri ®[1,2] ✉, Mark J. Buckley[3] & Keiji Tanaka[2]

Intra-individual behavioral variability is significantly heightened by aging or neuropsychological disorders, however it is unknown which brain regions are causally linked to such variabilities. We examine response time (RT) variability in 21 macaque monkeys performing a rule-guided decision-making task. In monkeys with selective-bilateral lesions in the anterior cingulate cortex (ACC) or in the dorsolateral prefrontal cortex, cognitive flexibility is impaired, but the RT variability is significantly diminished. Bilateral lesions within the fronto-polar cortex or within the mid-dorsolateral prefrontal cortex, has no significant effect on cognitive flexibility or RT variability. In monkeys with lesions in the posterior cingulate cortex, the RT variability significantly increases without any deficit in cognitive flexibility. The effect of lesions in the orbito-frontal cortex (OFC) is unique in that it leads to deficits in cognitive flexibility and a significant increase in RT variability. Our findings indicate remarkable dissociations in contribution of frontal cortical regions to behavioral variability. They suggest that the altered variability in OFC-lesioned monkeys is related to deficits in assessing and accumulating evidence to inform a rule-guided decision, whereas in ACC-lesioned monkeys it results from a non-adaptive decrease in decision threshold and consequently immature impulsive responses.

In the context of cognitive tasks requiring rapid responding, the time taken to reach a decision and then select and execute an action, the response time (RT), reflects the efficiency of involved cognitive functions[1–5]. The RT significantly fluctuates across trials even in highly trained individuals[3–6]. Such intra-individual trial-by-trial RT fluctuations might represent instabilities in the executive (cognitive) control of ongoing tasks. The RT variability might reflect weaker control, and/or lapse of attention and higher susceptibility to error commission. Indeed, a larger RT variability has been associated with occasional reduction in control or attention[5,7–9]. Accumulated evidence indicates that the intra-individual RT variability is significantly exaggerated in aging[1,6,8,10], in neuropsychological disorders and patients afflicted with

brain injuries[4,11–15], and has even been shown to predict the all-cause mortality rate of the elderly[12]. Therefore, understanding the neural substrates and underlying mechanisms of intra-individual RT variability might bring critical insight into the pathophysiological processes that underpin lapses of attention and cognitive deficits that predispose afflicted people to accidents and social disadvantages.

In line with the hypothesis that alterations in executive control and attention might underlie the trial-by-trial RT variability, imaging studies in humans have shown that activity in some of the main nodes of the attention and executive control networks such as the anterior cingulate cortex (ACC), posterior parietal cortex and dorsolateral prefrontal cortex (DLPFC), shows associations with RT and its trial-by-

[1]Cognitive Neuroscience Laboratory, Department of Physiology and Neuroscience Program, Biomedicine Discovery Institute, Monash University, Clayton, VIC 3800, Australia. [2]RIKEN Center for Brain Science, Wako, Saitama 351-0198, Japan. [3]Department of Experimental Psychology, Oxford University, Oxford OX1 3UD, UK. ✉e-mail: Farshad.mansouri@monash.edu

trial variability[5,7,16]. However, imaging studies have reported different, and even opposite, types of blood oxygen level-dependent (BOLD) signal change (increase, decrease, or temporal shift) in association with heightened RT variability. The negative and positive correlations between BOLD signals and RT variability have been associated with inadequate (low) levels of activation, and the compensatory recruitment of brain regions involved in cognitive control, respectively[5,7,16–20]. These differences might reflect the dynamic nature of the executive control of goal-directed behavior in which the role of neural circuits evolves in the course of consecutive trials: they might be involved in a kind of 'preparatory set' before the start of the trial[9], recruited by various cognitive functions within the trial, and also support post-trial learning and compensatory processes[5,16,21]. The recruited neural processes and the level of their involvement might also differ depending on the task context e.g., the currently relevant rule or the decision outcome (correct or error)[22–27]. Such dynamic changes might not be distinguished by the temporal resolution of current functional imaging techniques.

Although imaging studies in humans indicate a link between neural activity in prefrontal and medial frontal regions and trial-by-trial alterations in RT[1–3,5,7,10,14,16,19,20,28–31], they do not necessarily establish causality between particular brain regions and such variabilities in the context of cognitive tasks. Neuropsychological examinations of patients with various neurodevelopmental and neuropsychological disorders or brain damage have brought important insights regarding the link between the integrity of certain brain regions and intra-individual RT variability[2–4,8,13], however due to heterogeneity and inconsistency of lesion extent in patients and unilateral lesions in many cases, it has been difficult to delineate the causal role of particular prefrontal or medial frontal regions in intra-individual RT variability. Therefore, it is necessary to examine how selective and bilateral lesions (regional malfunction) in a particular brain region affect intra-individual RT variability in more highly controlled conditions. Macaque monkeys are suitable models for such an experiment, given the close similarity in the structure and organization of prefrontal and medial frontal regions between macaques and humans[32–35] and the fact that macaque monkeys can learn challenging cognitive tasks and perform hundreds of trials, which would enable reliable assessment of the intra-individual RT variability in a testing session[36–38]. Moreover, unlike in patients with different lesions, groups of macaques with different lesions may have broadly similar pre-operative experience with behavioral tasks facilitating cross-group comparisons.

Dominant models of decision-making[39–42] propose that the RT reflects three main processes: (1) evidence accumulation for a particular choice, which is reflected in the drift rate toward the decision threshold; (2) the decision threshold, which determines when evidence accumulation process ends and a response (rule-guided action) is delivered and (3) perceptual- and motor-related processes[39,41,43,44]. In the context of the Wisconsin Card Sorting Test (WCST), the available evidence for the relevant rule changes trial by trial because of the imposed uncued rule changes as well as the decision feedback (reward and error-signal/no-reward given after correct and error trials, respectively). When evidence for the relevant rule is high, the accumulation of evidence (drift to the threshold) will occur faster and therefore the RT will be shorter. However, when evidence for the relevant rule is low, accumulation of evidence will necessarily take longer and lead to a longer RT. In this context, executive control would normally enhance the efficiency of evidence accumulation (increase the slope of drift), but its impairment/instability would disrupt evidence accumulation and appear as a longer RT and more errors. In parallel, alterations in decision threshold might also affect the RT so that an abnormally lower threshold will terminate evidence accumulation and lead to shorter RT but also to higher error likelihood.

Psychophysical, functional imaging and modeling studies in humans suggest that both ACC and DLPFC are main nodes of an executive control network that supports evidence accumulation in the decision-making process[39,41,43,44]. The involvement of this executive control network, and particularly ACC, in controlling impulsive responses (actions) have also been reported[39,41,43]. Related models propose that impulsive responses result from abnormal decrease in decision threshold, which might lead to a premature response before sufficient evidence accumulation[39,41,43]. This threshold change would manifest as a shorter RT but higher error likelihood[39,41,43].

To establish the causal relationship between specific brain regions and intra-individual RT variability, we made selective bilateral lesions in six distinct prefrontal and medial cortical regions in macaque monkeys performing a WCST analog, which is a challenging rule-guided decision-making task demanding cognitive flexibility to deal with frequently shifting rules[25,38,45,46]. In our WCST analog (henceforth referred to as WCST) (Fig. 1a), the sample, the test items and their position were randomly changed trial-by-trial and there was no cue to the relevant rule or its frequent changes. Therefore, the monkeys had to find the relevant rule by trial and error and attain the rule-shift criterion (85% correct in 20 consecutive trials) with each rule. Considering this task design and the fact that control monkeys could shift between rules (attain high performance following each rule shift) by committing a limited number of errors[25], it is very unlikely that monkeys implemented an association-based strategy to adapt to frequent rule shifts[33]. In addition, in earlier studies we showed that monkeys could generalize the rule to novel items[38], which indicated they were applying a rule-based, but not association-based, strategy for action selection in the WCST. Previous lesion-behavioral studies with non-human primate models, in the context of the WCST analogs or set-shifting tasks, have shown the dissociable involvement of lateral, medial and orbital prefrontal regions in supporting cognitive flexibility in adapting to changing environments[25,47–50]. Neuronal activity recording and neuroimaging studies have shown that neural activities in a distributed network, including DLPFC, ACC, orbitofrontal cortex (OFC) and striatum, convey detailed information regarding rules and other task-related events in the WCST and set-shifting tasks[45,46,50–52].

In the current study, selective lesions were made within DLPFC, ACC, or superior part of the dorsal-lateral prefrontal cortex (Fig. 1b) (Table 1). These brain regions are critical nodes of the attention network in humans[24,26,53,54]. We also made selective bilateral lesions in the OFC (Fig. 1b) because recent studies suggest that OFC might play crucial roles in the executive control of rule-guided behavior in the WCST and other set-shifting tasks[25,45,48,55,56]. In humans, posterior cingulate cortex (PCC) and frontopolar cortex comprise the main nodes of the default mode network. Resting state functional connectivity studies have suggested the presence of a comparable network in non-human primates, although the functional homology remains to be established[57–60]. Imaging studies in humans have shown correlated activity change in these distributed brain networks in relation to the RT variability[5,10,16,18,19,28,30,61,62]. In this study, we also examined how selective lesions in OFC, frontopolar cortex and PCC affect the behavioral variability.

Previous studies, in the context of WCST and other set-shifting tasks, have indicated that after rule or set shifts, macaque monkeys efficiently learn to adapt to the rule change by applying the currently relevant rule, however they continue to commit occasional errors indicating that despite long-term training with task switching, interference of the irrelevant rule affects their rule-based action selection[25,38,63,64]. Therefore, we examined RT variability in correct (applying the relevant rule) or perseverative error (applying the irrelevant rule) trials, separately.

## Results

In each daily testing session, the monkeys performed 300 trials and, with a shift criterion of 85% correct in 20 trials, they could attain the rule-shift criterion maximally 15 times in a daily session. Therefore, the

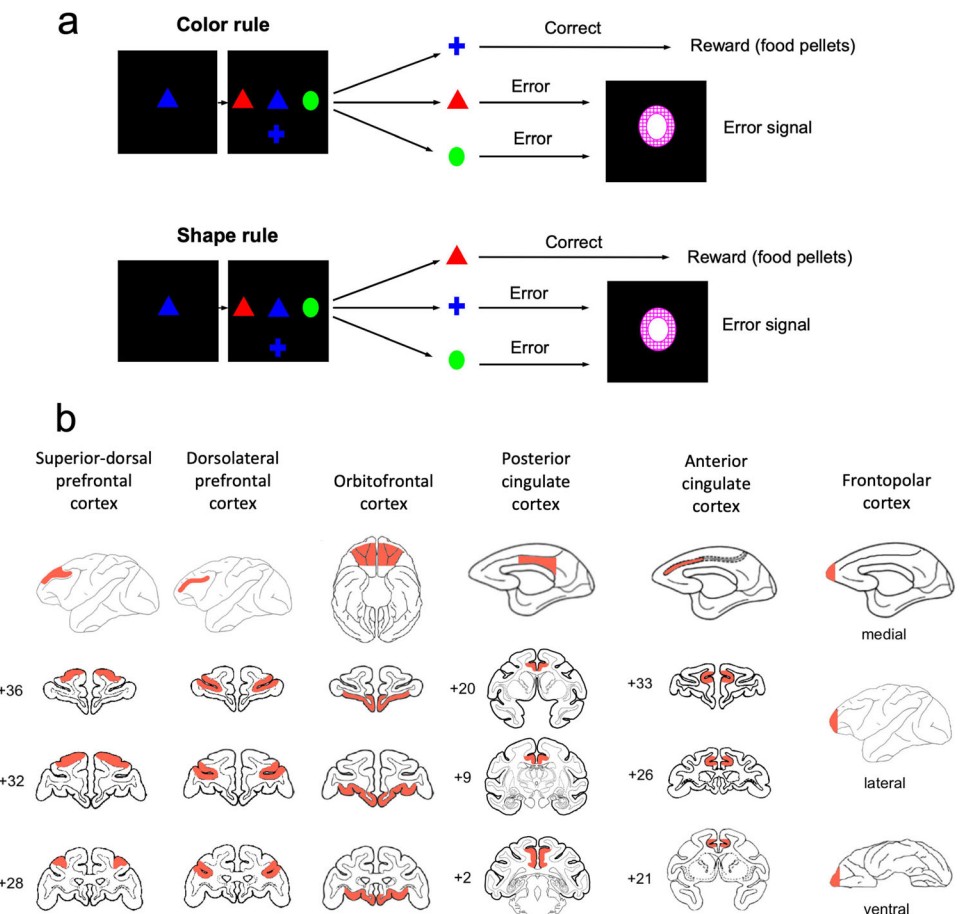

**Fig. 1 | Computerized versions of the WCST and intended lesion extent. a** In the Wisconsin Card Sorting Test (WCST), each trial commenced with sample onset at the center of the screen and after monkeys touched the sample and released their hand, three test items were presented on the left, right and bottom of the sample. A correct application of the matching rule led to the arrival of a reward. If monkeys did not match based on the currently relevant rule (touched one of the non-matching items) or did not respond within the response window, an error signal was presented and no reward was delivered. **b** The schematic diagrams show the extent of intended lesions in different groups of monkeys. Red regions show the extent of intended lesion (all lesions were bilateral). The details of lesion extent in each group are explained in the online Methods section. The lesion extents were largely as planned as assessed by microscopic inspection of post-mortem histological sections (see Supplementary Information for figures and discussions of lesion extent in individual animals) in all groups (except for in the posterior cingulate cortex group wherein coronal magnetic resonance images were inspected). Numerals: distance in mm from the interaural plane.

total number of rule shifts in a daily session reflected the monkey's cognitive flexibility in the day. To control the context in which decisions are made (i.e., the history of decision outcome: correct and error in preceding trials) we focused on correct trials that were preceded by correct trials (cC trials). We then calculated 'coefficient of variability' (Standard deviation/mean) for RT (RT-COV) in the second trial of each cC trial sequence (upper case letter). In perseverative error trials, the monkeys applied the irrelevant (previously relevant) rule. We selected perseverative error trials that were preceded by correct trials (cE trials) and calculated RT-COV in the second trial of each cE sequence.

### Consequence of selective lesions in ACC

When a nested ANOVA [Lesion-group (ACC/Control, between-subject factor) × Monkey (10 monkeys, between-subject factor) nested within Lesion-group] was applied to RT-COV in correct (cC) trials in ACC-lesioned and Control monkeys, there was a highly significant effect of Lesion-group (Table 2a): the RT-COV was significantly smaller in the ACC-lesioned group (Fig. 2a, b), which was in contrast to the consequence of lesions (larger RT-COV) in the PCC-lesioned monkeys (Fig. 3e). Applying the ANOVA to SD values led to the same conclusion (Table 2c and Fig S1a). Despite a lower RT variability, ACC-lesioned monkeys were significantly impaired in cognitive flexibility: compared to control monkeys, ACC-lesioned monkeys achieved significantly

lower number of rule-shifts per daily session (Table 2h), as has been reported in our previous studies[25].

When a two-way ANOVA [Response-type (cC/cE, within-subject factor) × Monkey (6 monkeys)] was applied to RT-COV in cC and cE trials of the Control group, the main effect of Response-type was highly significant (F(1, 84) = 61.05; $P < 0.001$, ηp2 = 0.42): RT-COV was larger in cE (error) trials. Then, we examined the effects of the ACC lesion on the RT variability difference between cC and cE trials. When a two-way nested ANOVA [Lesion-group (ACC/Control) × Response-type (cC/cE, within-subject factor) × Monkey (10 monkeys, nested within Lesion-group)] was applied to RT-COV in cC and cE trials, there was a significant interaction between the Response-type and Lesion-group factors (Table 2e). The difference between cC and cE trials was smaller in ACC-lesioned monkeys, which was mainly due to a reduced RT-COV in cE trials (Fig. 4a). The results were consistent when we applied the ANOVA to SD values (Table 2f and Fig. S4a).

### Consequence of selective lesions in DLPFC

When the nested ANOVA was applied to RT-COV in cC trials in DLPFC-lesioned and Control monkeys, the effect of Lesion-group was marginally significant (Table 2a): the RT-COV was smaller in the DLPFC-lesioned group (Fig. 2c, d). Although the effect of Lesion-group was highly significant with SD values: SD was smaller in the DLPFC-lesioned

**Table 1 | Demographic information about the two cohorts of monkeys in this study**

|  | Control[25] | ACC[25] | DLPFC[25] | OFC[25] | sdlPFC[25] | Frontopolar[65] | PCC[65] |
|---|---|---|---|---|---|---|---|
| First cohort | Yes | Yes | Yes | Yes | Yes | - | - |
| Second cohort | - | - | - | - | - | Yes | Yes |
| Macaca fuscata (Exact DOB is not available) (mean age at the time of operation) | 3 monkeys (7.3 years) | 2 monkeys (7 years) | 2 monkeys (8.5 years) | 1 monkey (9 years) | 2 monkeys (8 years) | - | - |
| Macaca mulatta (Age at the time of operation) | 3 monkeys (Age: 8, 8 and 8 years) | 2 monkeys (Age: 7 and 7 years) | 2 monkeys (Age: 7 and 7 years) | 2 monkeys (Age: 9 and 8 years) | 1 monkey (Age: 9 years) | 4 monkeys (Age: 7.5, 7, 6.75 and 7 years) | 3 monkeys (Age: 8, 9 and 8.2 years) |
| Male | 6 | 4 | 3 | 3 | 3 | 4 | 3 |
| Female | - | - | 1 | - | - | - | - |
| Trained and operated at RIKEN institute | 3 | 2 | 2 | 1 | 2 | 4 | 3 |
| Trained and operated at Oxford University | 3 | 2 | 2 | 2 | 1 | - | - |

(1) The first cohort included 6 adult monkeys in the Control (no lesion), 4 monkeys in the anterior cingulate cortex (ACC) group and 4 monkeys in the dorsolateral prefrontal cortex (DLPC) group. The 6 Control monkeys were then assigned to the orbitofrontal cortex (OFC: 3 monkeys) and superior dorsal-lateral prefrontal (sdlPFC: 3 monkeys) groups. All the monkeys, who were trained at Oxford University were *macaca mulatta*. (2) The second cohort of monkeys were all trained at RIKEN institute and included 4 adult monkeys in the frontopolar cortex (Frontopolar) group and 3 monkeys in the posterior cingulate cortex (PCC) group. The exact date of birth (DOB) was not available for one of the monkeys in the frontopolar group and his age was estimated based on his transfer to the experimental facility. The age has been mentioned at the time of surgery. It took, in average, 1.5 years to train each monkey to perform the WCST and collect the pre-lesion data. Therefore, monkeys' training started about 1.5 year before the age mentioned in this Table. *DLPFC* Dorsolateral prefrontal cortex, *ACC* anterior cingulate cortex, *OFC* orbitofrontal cortex, *sdlPFC* superior dorsal-lateral prefrontal cortex, *PCC* posterior cingulate cortex.

group (Table 2c and Fig. S1b), we have mainly considered the conclusion obtained with RT-COV (marginally significant), because both the RT variability and mean RT decreased in this case (see Methods, Data analyses). Despite a lower RT-variability, DLPFC-lesioned monkeys showed significant impairment in adapting to rule changes (Table 2h)[25]. Regarding the difference in fluctuations between cC and cE trials; there was no significant interaction between the Response-type and Lesion-group factors with RT-COV (Table 2e and Fig. 4b), while the interaction was significant with SD (Table 2f and Fig. S4b).

### Consequence of selective lesions in OFC
To assess the effects of OFC lesion, we applied a two-way ANOVA [Lesion (pre-lesion/post-lesion, within-subject factor) x Monkey (3 monkeys, within subject factor)] to RT-COV of correct responses in the OFC group. There was a highly significant main effect of Lesion factor (Table 2a) without significant Lesion x Monkey interaction (Table 2b): the RT-COV in cC trials became larger after OFC lesions (Fig. 2e, f). Applying the ANOVA to SD values also led to the same conclusion (Tables 2c and 2d and Fig. S1c). This result indicates that OFC-lesion led to a remarkable increase in RT variability. There was a significant impairment in cognitive flexibility after the OFC lesions (Table 2h)[25]. As for the difference in RT fluctuations between cC and cE trials, while the interaction between Response-type and Lesion factors was significant with RT-COV (Table 2e and Fig. 4c), there was no significant interaction with SD (Table 2f and Fig. S4c).

### Consequence of selective lesions in sdlPFC
When the two-way ANOVA was applied to RT-COV of correct responses in the sdlPFC-lesioned group, there was no significant main effect of Lesion factor (Table 2a and Fig. 3a). Applying the ANOVA to SD values also led to the same conclusion (Table 2c and Fig. S1d). The sdlPFC-lesioned monkeys did not show any deficit in shifting between rules, either (Table 2h)[25]. As for the difference in fluctuations between cC and cE trials, there was no significant interaction between Response-type and Lesion factors with RT-COV (Table 2e and Fig. 4d), while the interaction was marginally significant with SD (Table 2f and Fig. S4d).

### Consequence of selective lesions in the frontopolar cortex
When the two-way ANOVA was applied to RT-COV of correct responses in the frontopolar-lesioned group, there was no significant main effect

of Lesion-group (Table 2a and Fig. 3c). Applying the ANOVA to SD values also led to the same conclusion (Table 2c and Fig. S1e). Lesions within the frontopolar cortex did not affect monkey's ability in shifting between rules either (Table 2h)[65]. As for the difference in fluctuations between cC and cE trials, there was no significant interaction between Response-type and Lesion factors either with RT-COV (Table 2e and Fig. 4e) or SD (Table 2f and Fig. S4e).

### Consequence of selective lesions in PCC
When the two-way ANOVA was applied to RT-COV of correct responses in PCC-lesioned group, there was a highly significant main effect of Lesion (Table 2a) without a significant interaction (Table 2b): the RT-COV became significantly larger after PCC lesions (Fig. 3e). With SD values, the main effect of Lesion was significant but the interaction was also significant (Tables 2c, and 2d; and Fig. S1f). Although RT-COV was significantly increased in PCC-lesioned monkeys, they did not show any significant deficit in cognitive flexibility in shifting between rules (Table 2h)[65]. As for the difference in fluctuations between cC and cE trials, there was no significant interactions between Response-type and Lesion factors either with RT = COV (Table 2e and Fig. 4f) or SD (Table 2f and Fig. S4f).

Figure 5 summarizes the consequence of lesions on monkeys' RT variability and cognitive flexibility (number of rule-shifts) for all lesion groups.

### Response time at different levels of evidence for rule-guided actions
To help infer the underlying mechanisms of RT variability, we examined how the RT changed as the monkeys made multiple correct selections in consecutive trials. We classified correct trials, according to the number of correct trials preceding the current trial: The classification included eC (correct trial immediately after an error trial), ecC (a correct trial preceded by one correct trial after an error trial) and eccC (correct trial preceded by two consecutive correct trials after an error trial) trials. We hypothesized that monkeys' RT will be the longest in eC trials, when the lowest level of evidence exists to guide rule-based action selection, however monkeys' RT would be shorter in ecC and eccC trials because of accumulated evidence (receiving a reward for a correct selection of rule) in these trials. We found that in the Control monkeys the RT was the longest in eC trials, decreased in ecC trials,

**Table 2 | Summary of behavioral changes following selective lesions**

| | Measure | ANOVA structure | Effect type | ACC | DLPFC | ANOVA structure | Effect type | OFC | sdlPFC | Frontopolar | PCC |
|---|---|---|---|---|---|---|---|---|---|---|---|
| a | RT variability in cC trials | RT-COV in each session | Lesion-group (lesion/control) × Monkey (nested in Lesion-group) | Lesion-group (main) | Decreased (P<0.001, F(1,140)=35.66, np2=0.20) | Decreased (P=0.041, F(1,140)=4.26, np2=0.03) | Lesion (pre/post) × Monkey | Lesion (main) | Increased (P<0.001, F(1,42)=12.56, np2=0.23) | No change (P=0.33, F(1,42)=0.96, np2=0.02) | No change (P=0.78, F(1,56)=0.082, np2=0.001) | Increased (P<0.001, F(1,42)=30.31, np2=0.42) |
| b | | | | | | | | Lesion × Monkey (interaction) | P=0.68, F(2,42)=0.39, np2=0.02 | P=0.22, F(2,42)=1.55, np2=0.07 | P=0.32, F(3,56)=1.19, np2=0.06 | P=0.42, F(2,42)=0.90, np2=0.04 |
| c | | SD of RT in each session | | Lesion-group (main) | Decreased (P<0.001, F(1,140)=77.19, np2=0.36) | Decreased (P<0.001, F(1,140)=44.79, np2=0.24) | | Lesion (main) | Increased (P<0.001, F(1,42)=31.87, np2=0.43) | No change (P=0.42, F(1,42)=0.65, np2=0.02) | No change (P=0.87, F(1,56)=0.026, np2=0.001) | Increased (P<0.001, F(1,42)=25.64, np2=0.38) |
| d | | | | | | | | Lesion × Monkey (interaction) | P=0.65, F(2,42)=0.44, np2=0.02 | P=0.23, F(2,42)=1.52, np2=0.07 | P=0.43, F(2,42)=0.94, np2=0.05 | P=0.006, F(2,42)=5.72, np2=0.21 |
| e | RT variability difference between cC and cE trials | RT-COV in each session | Response-type (cC/cE) × Lesion-group (lesion/control) | Response-type × Lesion-group (interaction) | Decreased (P=0.041, F(1,140)=4.25, np2=0.03) | No change (P=0.24, F(1,140)=1.42, np2=0.01) | Response-type (cC/cE) × Lesion (pre/post) | Response-type × Lesion (interaction) | Decreased (P=0.008, F(1,42)=7.80, np2=0.16) | No change (P=0.80, F(1,42)=0.07, np2=0.002) | No change (P=0.055, F(1,56)=3.83, np2=0.064) | No change (P=0.20, F(1,42)=1.7, np2=0.04) |
| f | | SD of RT in each session | Response-type × Monkey (nested in Lesion-group) | Response-type × Lesion-group (interaction) | Decreased (P<0.001, F(1,140)=46.75, np2=0.25) | Decreased (P=0.015, F(1,140)=6.10, np2=0.04) | Monkey | Response-type × Lesion (interaction) | No change (P=0.46, F(1,42)=0.56, np2=0.01) | Decreased (P=0.040, F(1,42)=4.47, np2=0.10) | No change (P=0.16, F(1,56)=2.02, np2=0.04) | No change (P=0.38, F(1,42)=0.78, np2=0.02) |
| g | Mean RT in cC trials | | | | Decreased[25] | Decreased[25] | | | Increased[25] | Decreased[25] | No change[65] | No change[65] |
| h | Cognitive flexibility | | | | Decreased[25] | Decreased[25] | | | Decreased[25] | No change[25] | No change[65] | No change[65] |

"Decreased" and "Increased" indicate that the measure was significantly smaller and larger in the lesioned group than that in the control group (for ACC and DLPFC), respectively, or significantly smaller and larger in the post-lesion test than that in the pre-lesion test, respectively (for OFC, sdlPFC, FP, and PCC). "No change" means that there was no significant difference. The RT-COV or SD of RT in each of 15 daily sessions were used for analyses. For OFC, sdlPFC, Frontopolar and PCC, data were collected in 15 daily sessions for each of pre-lesion and post-lesion testing. *DLPFC* Dorsolateral prefrontal cortex, *ACC* anterior cingulate cortex, *OFC* orbitofrontal cortex, *sdlPFC* superior dorsal-lateral prefrontal cortex, *PCC* posterior cingulate cortex.

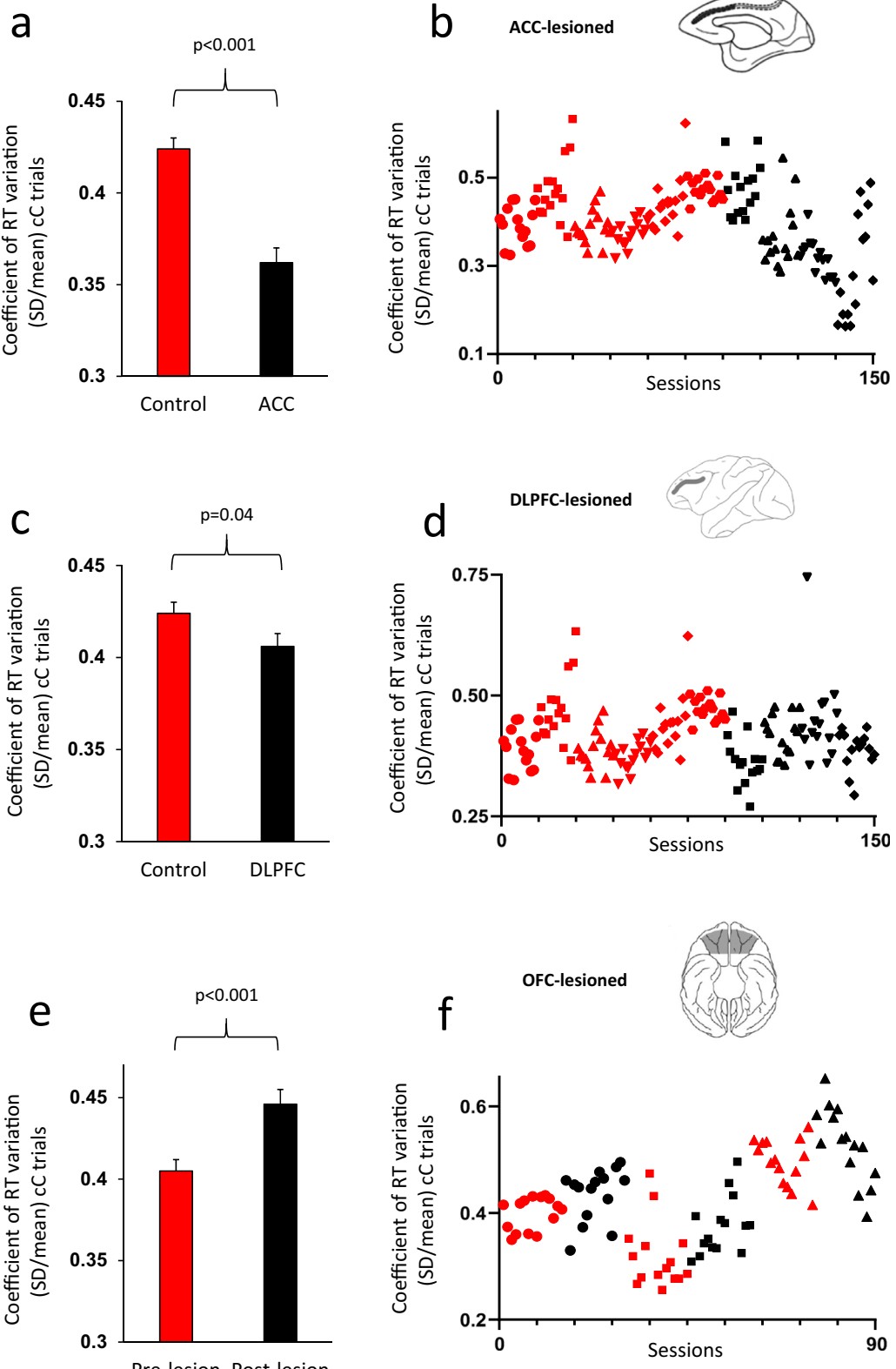

and further decreased in eccC trials (the black bars of Fig. 6a, b). We applied a multifactorial ANOVA [Evidence (eC/ecC/eccC, within-subject factor) × Monkey (6 Control monkeys)] to the mean RT in each session. The main effect of Evidence factor was highly significant (F(2168) = 945.15, $P < 0.001$, ηp2 = 0.92): the RT was significantly longer in eC trials. Pairwise $t$ tests (Bonferroni corrected) showed a significant

difference between eC and ecC ($p < 0.001$), between eC and eccC ($p < 0.001$), and between ecC and eccC trials ($p < 0.001$).

To assess the effects of ACC lesion, we applied a nested ANOVA (Table 3) to the mean RT in each session. The main effect of Lesion-group factor was highly significant (Table 3a): the RT was significantly shorter in the ACC-lesioned group compared to the Control group.

**Fig. 2 | Selective lesions in the DLPFC, ACC or OFC modulate intra-individual response time (RT) variability. a** The coefficient of response time variability (RT-COV) was significantly smaller in the ACC-lesioned monkeys, compared to Control group (F(1140) = 35.66). **b** The RT-COV of individual monkeys in 15 post-lesion sessions is shown for Control (red color) and ACC-lesioned monkeys (black color). The values for each monkey appear with a distinct marker shape. **c** The RT-COV was significantly smaller in the DLPFC-lesioned monkeys, compared to Control group (F(1140) = 4.26). **d** The RT-COV of individual monkeys in 15 post-lesion sessions is shown for Control (red color) and DLPFC-lesioned monkeys (black color). **e** The RT-COV is shown in the pre-lesion and post-lesion testing for the OFC-lesioned monkeys. RT-COV became significantly larger in the OFC-lesioned monkeys (F(1,42) = 12.56). **f** The RT-COV of individual monkeys in 15 pre-lesion (red color) and 15 post-lesion (black color) sessions is shown for OFC-lesioned monkeys. The cC sequence corresponds to a correct trial preceded by another correct trial. Data are presented as mean values ± SEM. The *p* value shows the main effect of Lesion factor in the ANOVA. All comparisons were two-sided. Dorsolateral prefrontal cortex (DLPFC), anterior cingulate cortex (ACC), orbitofrontal cortex (OFC).

The main effect of Evidence factor was also highly significant (Table 3b): the RT was significantly longer in eC trials, but decreased in ecC and eccC trials (Fig. 6a: ACC). The interaction between Lesion-group and Evidence factors was also highly significant (Table 3c): the RT difference between Control and ACC-lesioned monkeys for eC was significantly larger than that for ecC or eccC trials (Fig. 6a: ACC). Similar results were also found for DLPFC-lesioned monkeys (Table 3 and Fig. 6b). The main effect of Evidence was significant in all lesion groups (Table 3b). The main effect of Lesion factor was significant in OFC-lesioned monkeys without a significant interaction between Lesion and Evidence factors (Table 3b, c), which indicates that RT was longer in the post-lesion testing at all evidence levels. The Lesion effect was also significant in sdlPFC monkeys without a significant interaction between Lesion and Evidence factors (Table 3b, c): RT was shorter at all evidence levels. The main effect of Lesion was significant in PCC-lesioned, but not in frontopolar-lesioned monkeys (Table 3a). The interaction between Lesion and Evidence factors was significant in both PCC-lesioned and frontopolar-lesioned monkeys (Table 3c).

### RT variability was linked to the accuracy of upcoming decisions

In our previous study[9], we found that RT in humans and monkeys was dependent on the accuracy of upcoming decision suggesting that trial-by-trial changes in RT reflects fluctuations in the state of executive control. To examine whether RT variability was linked to the accuracy in the following trial (upcoming decision), we classified trials as cCc and cCe trials (e = error; c = correct; cCc refers to three consecutive correct trials and cCe refers to an error preceded by two consecutive correct trials). We calculated RT-COV in the second trial of each sequence (which is denoted by upper case letter). We applied a repeated-measure ANOVA [Trial-sequence (cCc/cCe, within-subject factor) × Monkey (6 control monkeys)] to the RT-COV in each session. The main effect of Trial-sequence was significant (F(1,84) = 5.77, *p* = 0.019, ηp2 = 0.06): the RT-COV was higher in cCe trials. There was no significant interaction between Trial-sequence and Monkey factors (F(5, 84) = 0.78, *p* = 0.56, ηp2 = 0.045). The higher RT variability in cCe trials suggest that when the executive control state was at lower levels (i.e., the likelihood of errors was higher in the following trial), the RT variability was higher in correct trials. However, when we examined the effects of lesions in ACC, DLPFC, sdlPFC, Frontopolar or PCC, there was no significant interaction between Lesion and Trial-sequence factors in any lesion group (p > 0.35 for all lesion groups), which suggests that lesions in these brain regions did not affect the difference in RT variability between cCc and cCe sequences.

### Discussion

We report dissociations in the involvement of different prefrontal and medial cortical regions in intra-individual RT variability. Dominant models, mainly emerging from studies in humans, propose that alterations in executive control and attention might underlie trial-by-trial RT variability: these models have suggested that heightened RT variability reflects weaker control, which might lead to lapses of attention and deficit in the task performance[3,5,10,14,16,18,19,28,30,31,62]. Here, mapping causal links between specific brain regions and intra-individual behavioral variability, we found that RT variability was significantly decreased in ACC-lesioned and in DLPFC-lesioned monkeys; and also accompanied by significant deficits in cognitive flexibility[25] (Fig. 5). At first glance, these intriguing findings might appear contradictory to the predictions of models assuming direct associations between heightened RT variability and instability of executive control. However, our findings can be well explained in the broader context of decision-making processes in which RT reflects three main processes: (1) evidence accumulation for a particular choice, (2) the decision threshold, and (3) perceptual- and motor-related processes[39,41,43,44].

In the context of cognitive tasks, RT might reflect various cognitive, sensory-motor and motivational aspects of behavior[3,5,16]. In a changing and complex environment, such as the WCST, where accumulation of evidence for available choices require executive control, the fluctuation of executive control will affect the rate of evidence accumulation and consequently correlate with RT fluctuations. In addition, alterations in decision threshold will also affect the RT and its fluctuation. In control (non-lesioned) animals, without changes in task contingencies, the sensory-motor processes would remain stable and therefore RT will mainly be associated with the rate of evidence accumulation and the decision threshold. In the context of the WCST, with its frequent uncued rule shifts, successful ongoing rule-guided behavior requires accumulation of evidence for the relevant rule by assessment of behavioral outcome (reward and error/no-reward)[25,66], by remembrance of the outcome of previous trials[67], by holding the memory of the currently relevant rule in working memory[38,46], and by inhibiting the currently irrelevant rule[32,33,68,69]. These processes are supported by executive control and therefore evidence accumulation processes would be dependent on the efficiency of executive control. In normal conditions, the efficiency and stability of executive control would be associated with the reduction in trial-by-trial RT fluctuation[9] and with the ability of monkeys to shift between rules. In error trials, executive control might be weaker (unstable) and that would lead to a slower drift rate (Fig. S3) and longer RT (Fig. S5), disrupted links between RT and rule-guided behavior[9], and eventually an erroneous action selection (lack of accumulated evidence for a rule).

Our previous findings[25] indicate that basic perceptual- and motor-related processes remained intact in ACC-lesioned and DLPFC-lesioned monkeys because they did not show any impairment in control tasks where no shift in rules was required. Our findings (Fig. 6) suggest that the monkeys' RT was significantly affected by the level of evidence accumulated following feedback to their choices. We have also previously reported that in Control (intact) monkeys, performance (percentage of correct responses) dropped to around 50% (chance level) following an error trial, however monkeys' performance increased after the first correct (rewarded) trial and continued to improve following consecutive correct trials[25]. However, following lesions within ACC or DLPFC, monkeys' performance increased more slowly following correct trials suggesting that ACC-lesioned and DLPFC-lesioned monkeys had impaired abilities in learning from consecutive rewards[25,52]. These results suggest that the executive control was significantly impaired and consequently led to difficulties in evidence accumulation in ACC-lesioned (and DLPFC-lesioned) compared to Control monkeys. Figure 7a, b show the proposed scheme in which the accumulation of evidence is depicted as drifting lines (black and red lines in Control and ACC-lesioned monkeys, respectively) progressing toward the theoretical decision threshold (blue and red

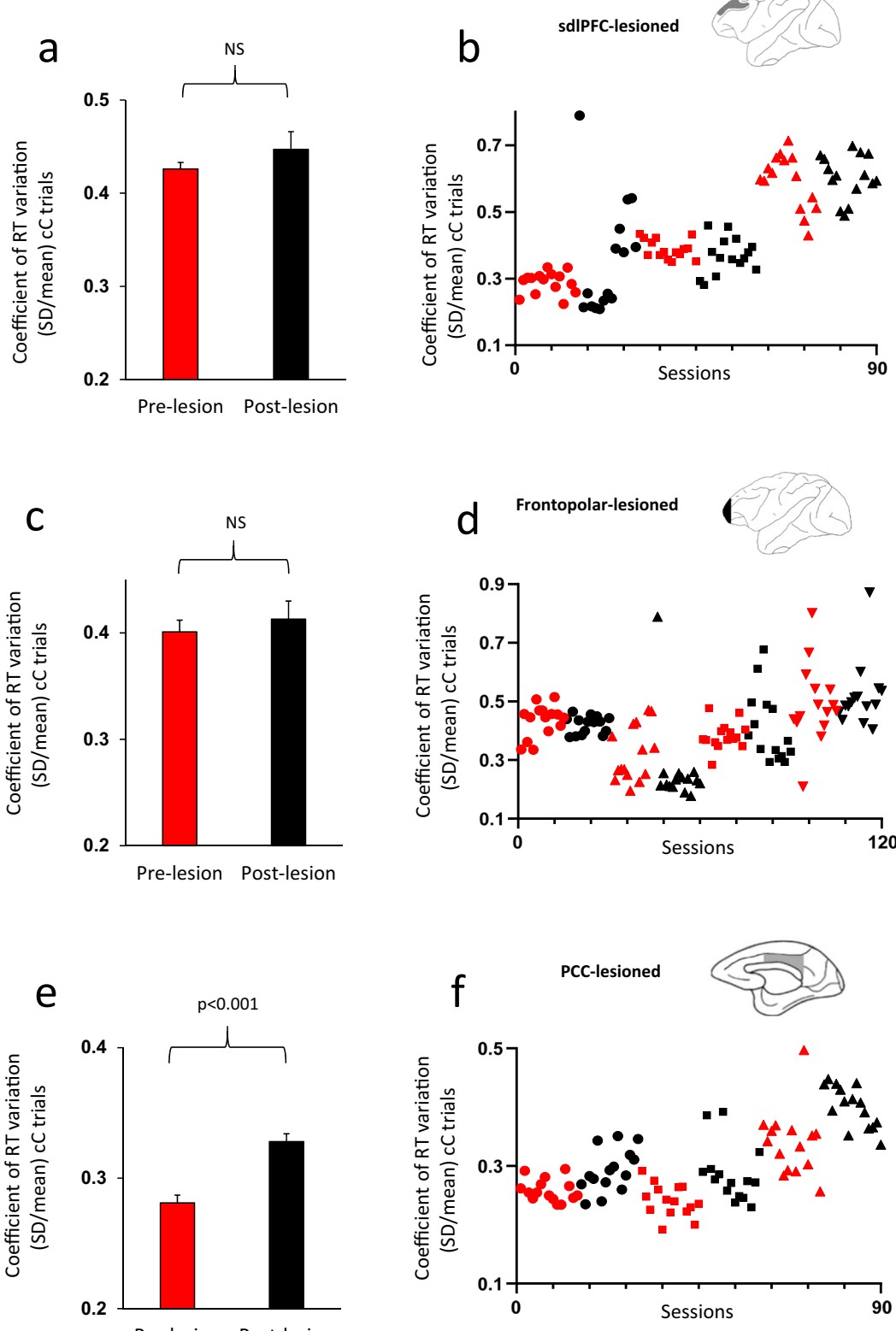

horizontal dotted lines in Control and ACC-lesioned monkeys, respectively). In the context of the WCST, when the available evidence for a rule is high (Fig. 7a), the process of evidence accumulation would proceed rapidly leading to a fast response (marked by vertical dashed lines); but when the available evidence for a rule is low (Fig. 7b), the evidence accumulation would require a longer time and therefore lead to a longer RT.

Importantly, we propose that the decision threshold was significantly lower in ACC-lesioned monkeys and probably DLPFC-lesioned monkeys as well (compared to Control monkeys) and consequently led to earlier termination of evidence accumulation and culmination in impulsive responses, which would also increase the error likelihood. Therefore, alterations in RT will be accompanied by more errors in the rule selection. A lower decision threshold also

**Fig. 3 | Consequence of selective lesions in the sdlPFC, frontopolar cortex or PCC on intra-individual behavioral variability. a** For sdlPFC-lesioned monkeys, there was no significant difference in RT-COV between the pre-lesion and post-lesion testing (F(1,42) = 0.96). **b** The RT-COV of individual monkeys in 15 pre-lesion (red color) and 15 post-lesion (black color) sessions is shown for sdlPFC-lesioned monkeys. The values for each monkey appear with a distinct marker shape. **c** For frontopolar-lesioned monkeys, there was no significant difference in RT-COV between the pre-lesion and post-lesion testing (F(1,56) = 0.082). **d** The RT-COV of individual monkeys in 15 pre-lesion (red color) and 15 post-lesion (black color)

sessions is shown for frontopolar-lesioned monkeys. **e** For PCC-lesioned monkeys, there was a significant difference in RT-COV between the pre-lesion and post-lesion testing, which appeared as a larger RT-COV (behavioral variability) following PCC lesion (F(1,42) = 30.31). **f** The RT-COV of individual monkeys in 15 pre-lesion (red color) and 15 post-lesion (black color) sessions is shown for PCC-lesioned monkeys. The *p* value and NS (Non-significant) indicate the main effect of Lesion factor in the ANOVA. Data are presented as mean values ± SEM. All comparisons were two-sided. Superior dorsal-lateral prefrontal cortex (sdlPFC), posterior cingulate cortex (PCC).

explains a reduction in RT variability. Figure 7 shows the rate of evidence accumulation at the highest (Fig. 7a) and the lowest (Fig. 7b) for Control and ACC-lesioned monkeys. In line with our findings (Fig. 6a); the RT was shorter in ACC-lesioned monkeys as compared with Control monkeys both at the highest and lowest levels of evidence. In addition, the RT variability (difference in RT between the highest and lowest levels of evidence) was smaller in ACC-lesioned monkeys as compared with Control monkeys. The distance between the two same-color vertical lines indicates the magnitude of RT difference within a session. The difference depicted as the red shadow for the ACC-lesioned group is smaller than the difference depicted as the blue shadow for the Control group (Fig. 7a, b). Thus, our proposed scheme predicts that, in the ACC-lesioned monkeys, the within-session RT variability would be significantly smaller compared to the Control monkeys because of the lower decision threshold (also see the Supplementary material). In fact, we found significant decrease in RT variability in ACC-lesioned (Fig. 2a) and marginally significant decrease in DLPFC-lesioned monkeys (Fig. 2c), which was accompanied by significant deficit in cognitive flexibility in the WCST[25].

In our conclusions regarding the decrease in RT variability and its underlying mechanisms, we have mainly focused on ACC-lesioned monkeys because the effects of ACC lesion were highly significant. We have previously reported that Control (intact) monkeys showed RT slowing in error trials (compared to the correct trials) and this error-slowing appeared regardless of the preceding trial (correct or error)[52]. The error slowing might reflect weaker control and the associated higher level of uncertainty in error trials[9]. However, error slowing was significantly attenuated in ACC-lesioned monkeys[52], which might be related to the lower decision threshold that led to a faster response despite lack of enough accumulated evidence.

We found that RT variability was significantly larger in error (cE) compared to correct (cC) trials (Fig. 4a). Figures S3a and S3b show evidence accumulation (two drift lines at two evidence levels) for Control and ACC-lesioned monkeys in correct and error trials, respectively. Our scheme predicts that the RT difference (the horizontal bidirectional arrow connecting the two vertical dashed lines) in correct trials (Fig. S3a) will be smaller than that in error trials (Fig. S3b) for both Control and ACC-lesioned monkeys (see the Supplementary material). The proposed scheme also predicts that the decrease in decision threshold, which might occur in ACC-lesioned monkeys, would lead to a smaller RT variability in ACC-lesioned monkeys in both correct (Fig. S3a) and error trials (Fig. S3b), which are supported by our observations (Fig. 4a). Our scheme also predicts that the lower decision threshold in the ACC-lesioned monkeys would lead to a smaller difference in RT variability between correct and error trials, as compared with the difference in the control monkeys. If the rate of evidence accumulation is constant over the time, namely, the drifting line is straight, the ratio of the magnitude of the RT variability is proportional to the ratio of the decision threshold for both correct (Fig. S3a) and error (Fig. S3b) trials (see the Supplementary material). In fact, our findings support this scheme by showing that the difference in RT variability between correct and error trials was significantly attenuated in ACC-lesioned monkeys (appeared as a significant interaction between Response-type and Lesion factors) (Fig. 4a).

The pattern of behavioral changes in the OFC-lesioned monkeys was unique among all lesion groups in that they showed a significantly higher level of RT variability (Fig. 2e), which was accompanied by remarkable deficits in cognitive flexibility (Table 2h)[25]. These indicate that the behavioral effects of OFC lesions were in stark contrast to the consequence of lesions in ACC suggesting that OFC plays a distinctly different role in RT variability, and presumably in trial-by-trial adjustment of executive control. In the frame of drifting model schema, we assume that the rate of evidence accumulation became lower after OFC lesion, at both the highest and lowest levels of evidence (Fig. 7c, d). In line with our findings (Fig. 6c); at both the highest and lowest levels of evidence, the RT was longer after OFC lesion (Fig. 7c, d). The reduced rate of evidence accumulation at various levels of evidence will also lead to higher RT variability (larger difference in RT between various levels of evidence) in OFC-lesioned monkeys (Fig. 7c, d, and the Supplementary material), which was what we observed (Fig. 2e). Different lines of evidence suggest that OFC is crucially involved in the executive control of rule-guided behavior in humans and monkeys[25,45,49,56,70–73]. Neuronal activity in OFC is also associated with monkeys' RT and accuracy in the upcoming decisions, which suggests that OFC might be involved in setting the executive control and restoring its allocation based on the behavioral outcome[9,72]. Importantly, OFC lesions disrupt behavioral improvement following reward, which suggests that OFC is crucially involved in evidence accumulation for the relevant rule by assessing the decision outcome[25,48]. In our proposed scheme, these contributions of OFC to executive control will enhance evidence accumulation for the relevant rule. Therefore, the significant increase in RT variability (Fig. 2e), and the deficit in cognitive flexibility[25], in OFC-lesioned monkeys, might be related to the impaired executive control and consequently lack of support for assessing and accumulating evidence for proper rule-guided decisions.

Having observed the effects of damage to the executive control network (such as ACC and DLPFC), we examined how selective lesions within frontopolar cortex or PCC affect RT variability. These two regions are main nodes of the default mode network in humans[61] and might also have corresponding roles in non-human primates[57–60]. Intriguingly, the effects of lesions differed between frontopolar cortex and PCC lesions in that RT variability was significantly increased in PCC-lesioned monkeys (Fig. 3e), but not in the frontopolar-lesioned monkeys (Fig. 3c). Lesions in frontopolar cortex or PCC did not affect monkeys' ability in shifting between rules (Table 2h)[65]. Although the role of default mode network and its neural substrate in non-human primates remain to be established[57–61], these findings bring insights regarding dissociations in contribution of the neural nodes, within the default node network and also between the executive control and default mode networks, to RT variability. There were also remarkable dissociations in contribution of anterior (ACC) and posterior (PCC) cortical regions of the cingulate sulcus to the trial-by-trial RT variability: lesions in ACC decreased RT variability (Fig. 2a) and the difference in RT variability between cC and cE trials (Fig. 4a), but PCC lesions increased RT variability (Fig. 3e) with no effect upon the difference between correct and error trials.

Intra-individual RT variability is predictive of individual's survival and has been considered as a behavioral biomarker of brain injury and

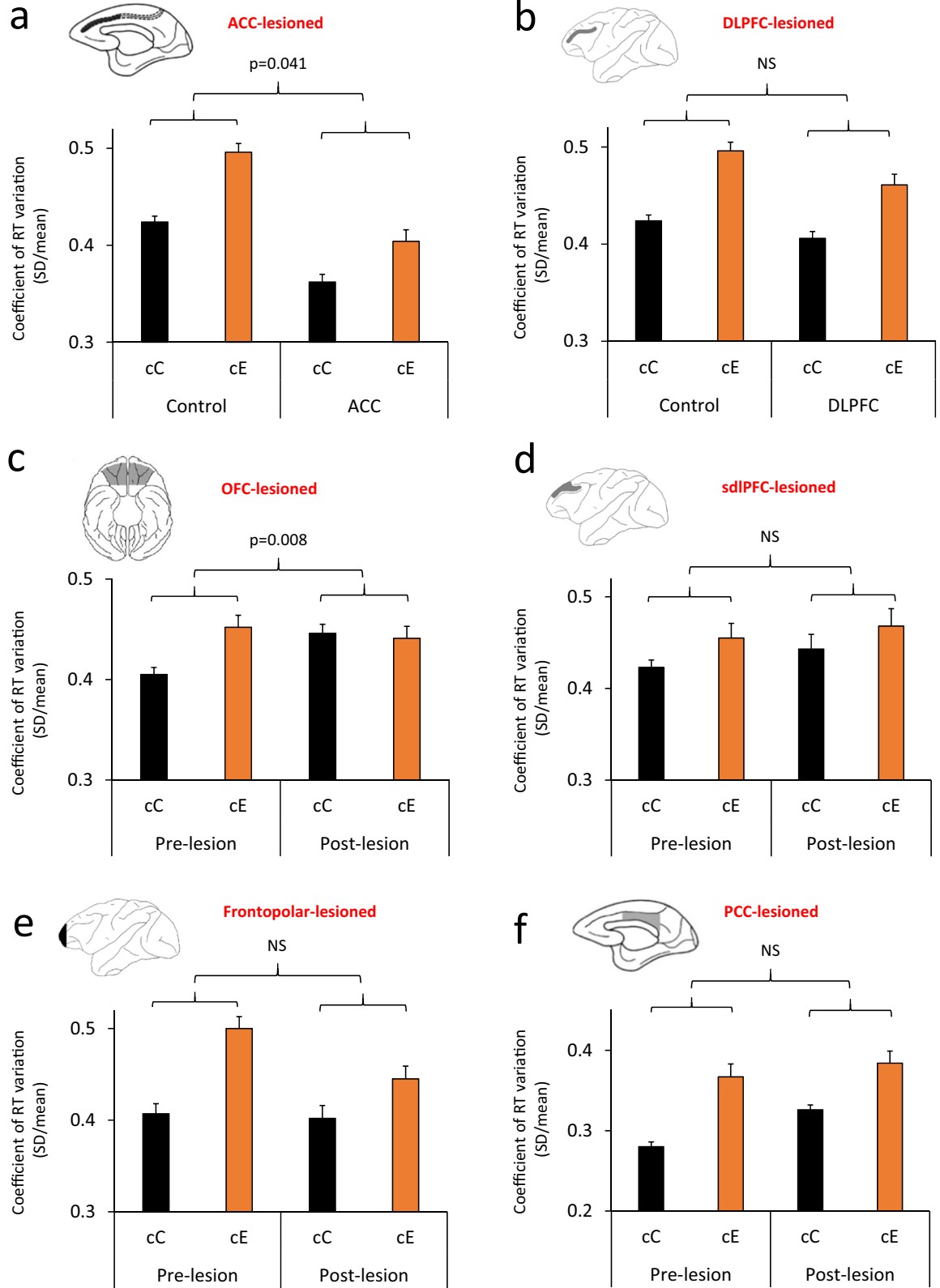

wide-ranging neuropsychological disorders; moreover, RT variability might even appear earlier than most other diagnosable symptoms. However, it has been difficult to link such changes to malfunctions of particular brain regions. Here, in an extensive lesion-behavioral study in macaque monkeys in the context of a challenging rule-shifting task, we mapped the causal link between various different brain regions and intra-individual RT variability and found remarkable functional

dissociations between the neural nodes of distributed executive control network (ACC, DLPFC, OFC), PCC and frontopolar cortex (Fig. 5 and Table 2). Our findings indicate that both extremes of RT variability (significant decrease or significant increase) might be associated with cognitive deficits in goal-directed behavior. The exaggerated RT variability in patients afflicted with traumatic brain injury or neuropsychological disorders might not be related to the selective

**Fig. 4 | Consequence of selective brain lesions on behavioral variability in error trials. a–f** The coefficient of response time variability (RT-COV) is shown in correct (cC) and error (cE) trials. **a** In ACC-lesioned monkeys, the RT-COV was decreased in both cC and cE trials, however the difference between cC and cE trials was significantly attenuated, which was mainly due to changes in cE trials (F(1140) = 4.25). **b** In DLPFC-lesioned monkeys, the RT-COV was decreased in both cC and cE trials, however the difference between cC and cE trials was not significantly affected (F(1140) = 1.42). **c** In OFC-lesioned monkeys, the RT-COV was significantly increased and the difference between cC and cE trials was significantly attenuated, however this was mainly due to changes in cC trials (F(1,42) = 7.80). **d** In sdlPFC-lesioned monkeys, there was no significant change in overall RT-COV or its difference between cC and cE trials (F(1,42) = 0.07). **e** In Frontopolar-lesioned monkeys, the RT-COV was significantly larger in error (cE) trials in both before and after the lesion, however the difference between cC and cE trials was not significantly affected (F(1,56) = 3.83). **f** In PCC-lesioned monkeys, the RT-COV was significantly larger in error trials in both before and after the lesion, however the difference between cC and cE trials was not affected (F(1,42) = 1.71). The *p* value shows the interaction between the Lesion and Response-type (cC/cE) factors in the ANOVA. Dorsolateral prefrontal cortex (DLPFC), anterior cingulate cortex (ACC), orbitofrontal cortex (OFC), superior dorsal-lateral prefrontal cortex (sdlPFC), posterior cingulate cortex (PCC). NS Non-significant. Data are presented as mean values ± SEM. All comparisons were two-sided.

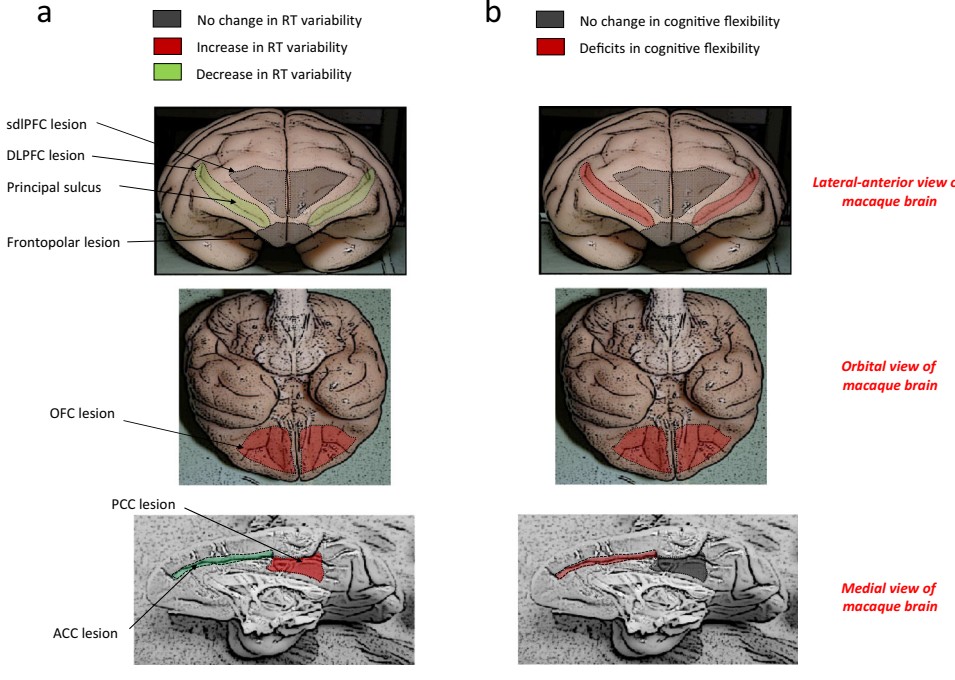

**Fig. 5 | Dissociations in the involvement of six cortical regions in intra-individual RT variability. a** The effects of the bilateral lesion in different brain areas on the RT variability. Red and green colors indicate significant effects, which appeared as increased and decreased RT-COV, respectively. Gray color indicates no significant effect. All lesions were bilateral; however, the lesion extent is shown only on one hemisphere for ACC and PCC. **b** The effects of the bilateral lesion in different areas on the cognitive flexibility. Red and gray colors indicate significant impairment and no deficits, respectively. Dorsolateral prefrontal cortex (DLPFC), anterior cingulate cortex (ACC), orbitofrontal cortex (OFC), superior dorsal-lateral prefrontal cortex (sdlPFC), posterior cingulate cortex (PCC).

malfunction of ACC because selective lesion in ACC was found to decrease RT variability. In addition, lesions in frontopolar cortex or sdlPFC do not affect RT variability. Selective lesions in OFC or in PCC led to significant increases in RT variability; however, only OFC-lesioned monkeys showed concomitant increase in RT variability and deficits in cognitive flexibility (Fig. 5).

RT variability is exaggerated in aging[8,10] and age-related changes in cognitive flexibility has been reported in humans and monkeys[74,75]. Early damage to prefrontal cortex or its broader network might also affect cognitive flexibility[76,77]. Our study examined RT variability and the effects of selective lesions in adult monkeys and found significant dissociations in contribution of frontal regions to RT variability. Future studies examining RT variability in aged non-human primate models might help to delineate the underlying mechanisms of age-related RT changes and concomitant cognitive declines. Furthermore, investigating the early-onset damage to prefrontal cortical regions might bring further insights regarding the developmental changes in RT variability and the contribution of prefrontal cortical regions[77,78]. Our findings in non-human primate models have great translatability to understand the neural basis of behavioral variability in humans, however direct generalization of our findings to human population needs

to be done cautiously considering that our study was conducted in a limited number of monkeys to minimize the animal use, and that a long-term training (about 1 year for each monkey) was required to learn performing the WCST.

We conclude that OFC has a unique contribution to RT variability and associated cognitive deficits, which might be related to its crucial role in assessing the behavioral outcome and adjusting evidence accumulation for making effective rule-guided decisions. This interpretation is also supported by previous studies in non-human primates indicating the crucial role of OFC in assessing the behavioral outcome and in shifting between rules and strategies[45,48,56,70–73,79]. Previous studies[25,32,80–82] have ascribed various critical functions for ACC in executive control of goal-directed behavior, such as monitoring demands (conflict, uncertainty) for allocation of control and assessing the outcome of actions. Our findings support the possibility of impairments in these functions, which would adversely affect evidence accumulation in the context of WCST; however, we propose that concomitant decrease in decision threshold in ACC-lesioned monkeys is the most parsimonious explanation of our findings, because it consistently explains the decrease in RT variability and other aspects of their behavior in the WCST. This scheme also conforms well with

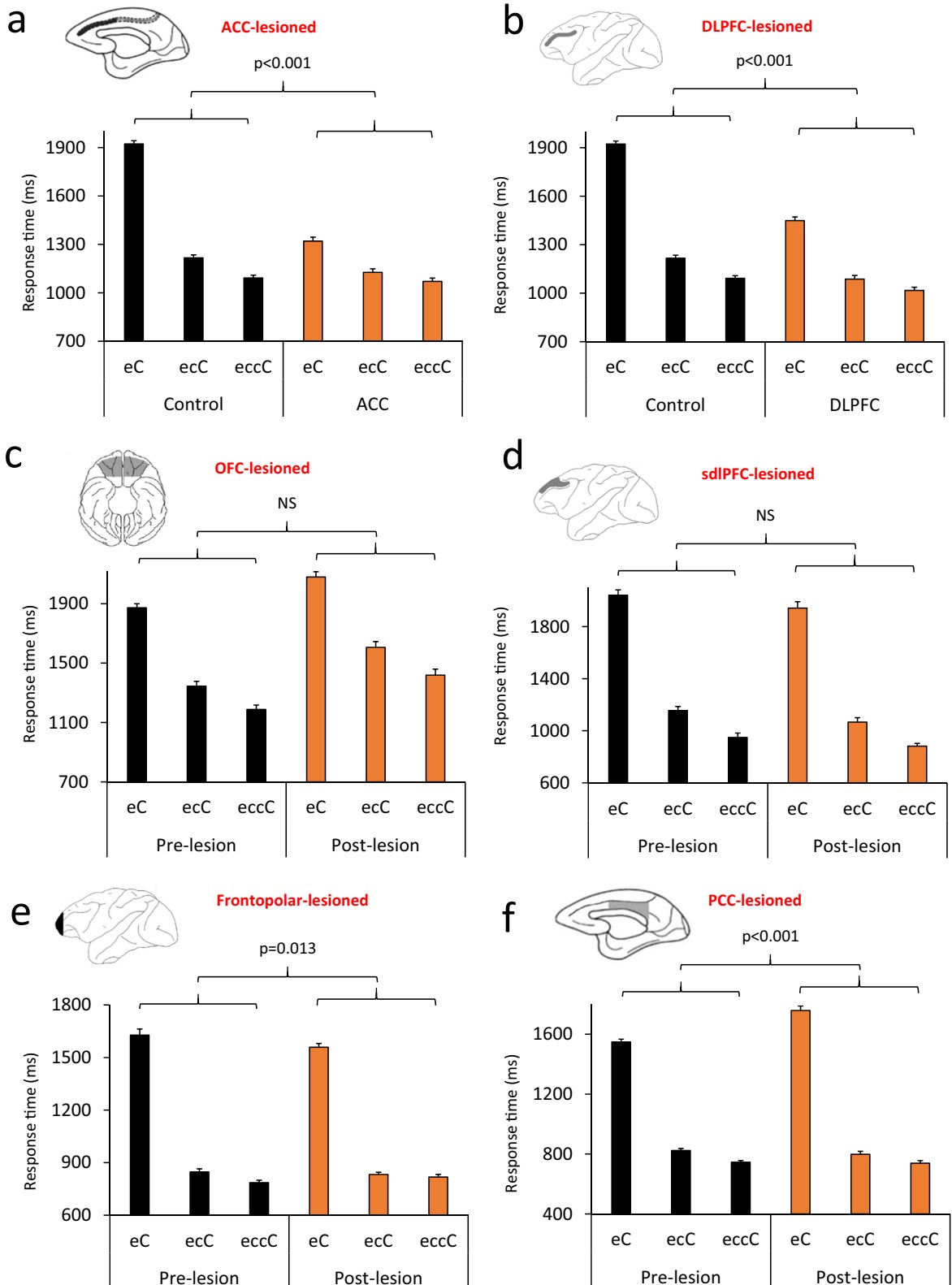

previous findings in humans showing that impulsive responses with higher error likelihood, as manifested in some neuropsychological disorders, might be linked to ACC dysfunction[41,83].

Previous studies[47,48,74,84], examining the consequence of lesions in monkeys' prefrontal cortex, have mainly focused on alterations in mean RT and accuracy (% correct/error), however our findings suggest that RT variability might be significantly and distinctively affected by malfunction of specific frontal cortical regions. Re-examining the RT variability in these studies might bring further insight to the underlying mechanisms of associated behavioral deficits. Our findings in monkeys take a significant step toward understanding the causal link between the function of particular brain regions and intra-individual RT variability in the context of goal-directed behavior and may bring critical insights to the neural substrate and pathophysiological mechanisms

**Fig. 6 | Response time (RT) at different levels of evidence for rule-guided actions. a** Monkeys' RT is shown in eC (a correct trial preceded by an error trial; e = error, C = correct), ecC and eccC trial sequences. RT was calculated in the current correct trial (upper case C) depending on the history (lower case letters). In Control monkeys, RT was the longest in eC trials, when the lowest level of evidence exists to guide rule-based action selection, however it was shorter in ecC and eccC trials. Compared to Control monkeys, ACC-lesioned monkeys had a shorter RT in all trial types (F(1140) = 105.69), however the difference in RT between Control and ACC-lesioned monkeys was the largest in eC trials (the lowest level of evidence). **b** A similar pattern of evidence-dependent modulation of RT was seen in the DLPFC-lesioned monkeys (F(1140) = 105.39). **c** Evidence-dependent modulation of RT was seen in OFC-lesioned monkeys, however their RT was longer at all evidence levels in the post-lesion testing (F(1,42) = 33.32). **d** Evidence-dependent modulation of RT was seen in sdlPFC-lesioned monkeys, however their RT was shorter at all evidence levels in the post-lesion testing (F(1,42) = 4.19). Evidence-dependent modulation of RT was seen in frontopolar-lesioned (F(1,56) = 0.86) (**e**) and PCC-lesioned monkeys (F(1,42) = 7.77) (**f**). The $p$ value shows the interaction between the Lesion and Evidence (eC/ecC/eccC) factors in the ANOVA. Data are presented as mean values ± SEM. All comparisons were two-sided. Dorsolateral prefrontal cortex (DLPFC), anterior cingulate cortex (ACC), orbitofrontal cortex (OFC), superior dorsal-lateral prefrontal cortex (sdlPFC), posterior cingulate cortex (PCC).

**Table 3 | Results of analyses for changes in Response time (RT) at different levels of evidence**

| | Measure | ANOVA structure | Effect type | ACC | DLPFC | ANOVA structure | Effect type | OFC | sdlPFC | Frontopolar | PCC |
|---|---|---|---|---|---|---|---|---|---|---|---|
| a | Mean RT in each session | Lesion-group (lesion/control)× Evidence (eC/ecC/eccC) × Monkey (nested in Lesion-group) | Lesion-group (main effect) | $P < 0.001$, F(1140) = 105.69, ηp2 = 0.43 | $P < 0.001$, F(1140) = 105.39, ηp2 = 0.43 | Lesion (pre/post) × Evidence (eC/ecC/eccC) × Monkey | Lesion (main effect) | $P < 0.001$, F(1,42) = 33.32, ηp2 = 0.44 | $P = 0.047$, F(1,42) = 4.19, ηp2 = 0.09 | $P = 0.36$, F(1,56) = 0.86, ηp2 = 0.02 | $P = 0.008$, F(1,42) = 7.77, ηp2 = 0.16 |
| b | | | Evidence (main effect) | $P < 0.001$, F(2,280) = 751.76, ηp2 = 0.84 | $P < 0.001$, F(2,280) = 1055.93, ηp2 = 0.88 | | Evidence (main effect) | $P < 0.001$, F(2,84) = 341.23, ηp2 = 0.89 | $P < 0.001$, F(2,84) = 780.56, ηp2 = 0.95 | $P < 0.001$, F(2,112) = 914.58, ηp2 = 0.94 | $P < 0.001$, F(2,84) = 1759.37, ηp2 = 0.989 |
| c | | | Lesion-group × Evidence (interaction) | $P < 0.001$, F(2,280) = 214.98, ηp2 = 0.61 | $P < 0.001$, F(2,280) = 103.22, ηp2 = 0.42 | | Lesion × Evidence (interaction) | $P = 0.65$, F(2,84) = 0.43, ηp2 = 0.01 | $P = 0.83$, F(2,84) = 0.19, ηp2 = 0.004 | $P = 0.013$, F(2,112) = 4.51, ηp2 = 0.08 | $P < 0.001$, F(2,84) = 59.08, ηp2 = 0.58 |

The mean RTs in eC (correct trial immediately after an error trial), ecC (a correct trial preceded by one correct trial after an error trial) and eccC (correct trial preceded by two consecutive correct trials after an error trial) trials in each daily session were used for analyses.

that underlie altered response variability and related cognitive deficits in patients afflicted with brain damage or neuropsychological disorders.

## Methods
### Study design and lesion groups
**Macaque model.** 21 macaque monkeys (7 *macaca fuscata* and 14 *macaca mulatta*) were trained to perform a computerized analog version of the WCST. Seven monkeys in the first cohort of animals were trained, operated and tested at Oxford University and the rest of studies were conducted at RIKEN institute. Table 1 includes demographic information (sex, species) of all 21 monkeys. The effects of lesions on the ability of ACC-lesioned[25,52], DLPFC-lesioned[25], OFC-lesioned[25,45], frontopolar-lesioned[65] and PCC-lesioned[65] monkeys in shifting between rules (cognitive flexibility) have been reported in our previous publications[25,42,50,63]. All experimental procedures in Japan conformed to the ethics guidelines specified by RIKEN Center for Brain Science. All experimental procedures at Oxford University followed the guidelines of the UK Animals (Scientific Procedures) Act of 1986, licensed through the UK Home Office, and approved by Oxford University Committee on Animal Care and Ethical Review.

In the first cohort of macaque monkeys, 14 monkeys were trained to perform a computerized version of the WCST (Fig. 1a). Then, based on individuals' pre-lesion performance (mean number of rule-shifts in each testing session), the monkeys were assigned to three separate groups of matched abilities. The range and mean of the numbers of pre-lesion shifts between rules were comparable between groups. In one group of 4 monkeys, bilateral lesions were made in both banks and fundus of principal sulcus on the lateral surface of prefrontal cortex (DLPFC group) (Figs. 1b and S6), in another group of 4 monkeys, bilateral lesions were made within ACC (ACC group) (Figs. 1b and S6), but the other 6 monkeys (Control group) did not receive any lesion and remained as unoperated controls. Monkeys' performance in the 15 post-lesion sessions was

compared between the lesioned and control groups. A two-week rest was considered after the lesion operation for all groups. Unoperated control (intact) monkeys also rested for 2 weeks between pre-lesion and post-lesion testing. In the second stage of the study, the 6 aforementioned Control monkeys were assigned to two performance-matched lesion groups. In three monkeys, bilateral lesions were made within the OFC (OFC group) (Figs. 1b and S7) and in the other three monkeys, bilateral lesions were made within superior part of the dorsal-lateral prefrontal cortex (sdlPFC group) (Figs 1b and S7). For the OFC and sdlPFC groups, the consequence of lesions was examined by comparing monkeys' behavior between 15 pre-lesion testing sessions and 15 post-lesion sessions. Monkeys in the OFC and sdlPFC groups had performed more sessions compared to the ACC and DLPFC groups. However, the effects of lesions within the OFC or in the sdlPFC on behavioral measures were assessed by comparing the pre-lesion and post-lesion performance (repeated-measure design). Therefore, the additional practice with the WCST, before making the lesions (while they served as the Control group), could not explain the effects of lesions in the OFC or sdlPFC.

In the follow up experiments with another cohort of monkeys, we trained 7 macaque monkeys to learn the WCST and then based on individuals' pre-lesion performance (mean number of rule-shifts in each testing session), the monkeys were assigned to two separate groups of matched abilities. One group received bilateral lesions in frontopolar cortex (frontopolar group, n = 4) (Fig. 1b and S6), and the other 3 monkeys did not receive any lesion (n = 3). In the second stage of this study, the three unoperated monkeys of the second cohort received bilateral selective lesions in the posterior cingulate cortex (PCC) (Fig. 1b and S7). For PCC-lesioned and frontopolar-lesioned animals, we compared RT variability between the pre-lesion and post-lesion performance. The 3 unoperated monkeys performed the WCST while the 15 pre- and 15 post-operative testing sessions were completed for the frontopolar-lesioned monkeys. After completion of the post-operative sessions in frontopolar-lesioned monkeys, the 3 unoperated

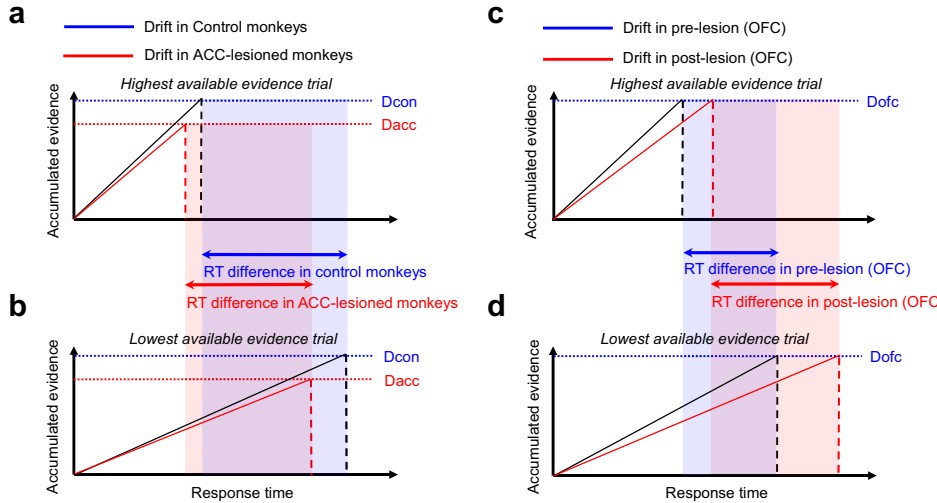

**Fig. 7 | Drifting model predicting the consequence of lesions in ACC, DLPFC and OFC on RT variability.** The two cases of drifting model performance in which the rate of evidence accumulation is the highest (**a**) and the lowest (**b**). The black and red oblique lines represent the drift rate for Control and ACC-lesioned monkeys, respectively. The explanation is given for ACC-lesioned monkeys, but can also be considered for the DLPFC-lesioned monkeys. The abscissae and ordinate represent the time and the amount of accumulated evidence for a particular response, respectively. The model assumes that the evidence for each response is accumulated constantly toward the decision threshold and a response is made when the accumulated evidence reaches the threshold. We assume that the decision threshold is significantly lower in the ACC-lesioned monkeys (D-ACC: red dotted line) as compared with that in the control monkeys (D-control: blue dotted line). The distance between the two same-color vertical lines indicates the magnitude of RT difference (bidirectional horizontal arrows) within a session. The difference depicted for the ACC-lesioned group (the light red region) is smaller than the difference for the Control group (the light blue region). This scheme is consistent with the results presented in Figs. 2a and 6a. **c** The black and red oblique lines represent the drift rate before and after OFC lesion, respectively when the rate of evidence accumulation is the highest (**c**) and the lowest (**d**). Decision threshold in OFC-lesioned monkeys (D-OFC) is shown with blue dotted line. We assume that the evidence accumulation is significantly impaired after OFC lesion, which would manifest as slower drift rates (red lines) at different levels of available evidence, compared to the pre-lesion state (black lines). The RT difference after OFC lesion (the light red region) is larger than the difference before OFC lesion (the light blue region). This scheme is consistent with the results presented in Figs. 2e, and 6c. Refer to Supplemental material: 'Computational background of Fig. 7' for the computational background. Dorsolateral prefrontal cortex (DLPFC), anterior cingulate cortex (ACC), orbitofrontal cortex (OFC).

animals received lesions in the PCC. Therefore, the 3 monkeys in the PCC group had more practice with the WCST (compared to the frontopolar-lesioned group). However, as mentioned above, the effects of lesions within the frontopolar cortex or in the PCC were assessed by comparing the pre-lesion and post-lesion performance and therefore, the additional practice with the WCST could not explain the effects of lesions in the frontopolar or PCC groups.

**Testing and training for cognitive tasks**
Monkeys were transferred to the experimental room by a transfer-testing cage and positioned in front of a touchscreen. Monkeys could freely move within the testing cage and perform the cognitive task. Open bars at the front of testing cage enabled accessing the touchscreen and a food box in which a food pellet was delivered for correct responses. Monkeys received their daily food in the experimental room. A computer-controlled food box, containing the daily food, was opened at the end of each training/testing session and monkeys were given enough time to access their daily food.

Monkeys performed a computerized version of the WCST with the color and shape rules (Fig. 1a). A set of 36 visual stimuli (made of six colors and six shapes) was used in the WCST. In each trial, a sample was selected and presented randomly, without replacement, until all 36 different samples were used and therefore none of the samples was repeated until the entire set was presented in consecutive trials. In each trial, the sample was shown at the center of the touchscreen and after the monkeys touched the sample, then three test items were presented surrounding the sample. One of the test items matched the sample in shape, another test item matched the sample in color and the other test item did not match the sample in either color or shape. Then, monkeys had to select and touch one of the test items that matched the sample according to the relevant rule (matching based on color when color rule was relevant; or matching based on shape when

the shape rule was relevant) within a limited response window (3000 ms). A banana-flavored food pellet (190 mg) was provided, as a reward, for each correct response. However, after an erroneous response a visual error signal was presented and no reward was given. In each block of trials, the monkeys had to reach a shift criterion of 17 correct in 20 consecutive trials (85% correct) and then the block changed (a new rule became relevant) without any notification. Monkeys were allowed to perform 300 trials in each daily session. The number of rule-shifts, percentage of corrects and mean response time in correct trials were calculated in each daily testing session. Additional details regarding the setup for training and testing in monkeys and the training steps for the WCST have been reported in our previous publications[25,45,46,85,86].

**Control behavioral tests**
In the post-lesion behavioral tests, we also included control tasks in which the rule (color- or shape-matching) remained constant within a daily session (no rule shift was required in the daily testing session). Performance of monkey in all groups (DLPFC, ACC, Control, OFC, sdlPFC, frontopolar and PCC) were comparable and at high level and no group showed any deficit in performing the control tasks. This indicates that monkeys' sensory, perceptual, motor and attentional abilities remained intact in all experimental groups[25].

**Surgery**
All surgeries were conducted in sterile conditions while monkeys were deeply anaesthetised. On the surgery day, the monkeys were sedated, intubated by a tracheal tube and then connected to an artificial respirator and remained anaesthetized with Isoflurane (1.0–3.0%) during surgery. The same neurosurgeon performed all surgeries at RIKEN institute and Oxford University. All aspiration lesions were visually guided using a surgical microscope. In order to access the

target brain region, a bone flap was raised over the left and right prefrontal cortex and then the dura was opened and reflected. Anatomical landmarks were examined to determine the extent of lesion in each animal based on pre-defined criteria. After exposing the brain regions, we used a small-gauge metal aspirator to carefully remove the cortex in the intended brain region. The aspirator was connected to a finely controlled suction system and insulated up to the tip to allow finely targeted electro-cautery. The same procedure was done in the left and right hemispheres. After completion of the lesions in each hemisphere, the dura mater was sewn back and the bone flap was re-positioned and stabilized by dissolvable sutures connected to the skull. The wound was closed and the skin was sewn back. For additional details regarding the surgical approach and pre-operative and post-operative procedures for making selective brain lesions please see our previous publications[25,86].

Inactivation studies using chemicals such as GABA agonizts (e.g., Muscimol) or neurotoxic compounds (e.g., Ibotenic acid) may allow localized inactivation or death of neurons at the injection sites, but cannot mimic the effects of complete disruption of large cortical regions, which were targeted in this study. Increasing the concentration or volume of injections for making complete lesions would cause certain involvement of nearby cortical areas, which would prevent proper interpretation of the lesion effects on cognitive functions. In addition, new molecular-genetic techniques for controlled inactivation/activation of neuronal populations (e.g., DREADD or optogenetics), which have been used in rodent models, are still being developed for primates[87,88]. The success rate in transferring the genetic codes to primate neurons is still low and therefore not feasible for complete and bilateral inactivation of deep and large cortical areas, which were targeted in this study. In this study, we used visually-guided aspiration lesion technique, which is still one of the most suitable and currently-available procedure in macaque monkeys to address the goal of this project. Although, the possibility of damage to the immediately underlying white matter cannot be ruled out, we made utmost care during surgery to avoid any deep damage to the underlying white matter and therefore, the possibility of any significant damage to major fascicles was very low.

### Intended extent of lesions within different brain regions

Supplementary figures S6 and S7 include the details of lesion extent for individual monkeys in each lesion group.

**Anterior cingulate sulcus (ACCs) lesion.** The extent of intended lesions in the ACC group (Figs. 1b, S6) covered the cortex in the dorsal and ventral banks and depth of the anterior cingulate sulcus, which correspond to cytoarchitectonic areas 24c, 24c'[35]. The posterior border of the lesion in the cingulate sulcus started at an imaginary line passing through the midpoint of the precentral dimple and the lesion extended anteriorly (rostrally) to include the entire extent of the cingulate sulcus. For the two out of the four ACC-lesioned monkeys, the lesions were complete and as intended, however in another ACC-lesioned monkey the lesion extent was larger than intended in one hemisphere, and in the other monkey, the lesion did not extend as far posteriorly as in the other three monkeys to avoid cutting the ascending branches of the anterior cerebral artery[25].

**Dorsolateral prefrontal cortex (DLPFC) lesion.** The extent of intended lesion in the DLPFC group (Figs. 1b, S6) covered the entire anterior-posterior extent of cortex in both banks and fundus of the principal sulcus. The lesion also extended to surrounding cortical regions 2–3 mm dorsal and ventral to the lips of the principal sulcus on the lateral surface of the prefrontal cortex. Therefore, the DLPFC lesion included the middle portion of cytoarchitectonic areas 46 and 9/46[35]. Histological examination in two monkeys and 3D structural MRI in the

two other monkeys confirmed that the lesion extent was as intended in all four DLPFC-lesioned monkeys[25].

**Orbitofrontal cortex (OFC) lesion.** The extent of intended lesions in the OFC group (Figs. 2b, S7) was limited laterally by the lateral orbital sulcus and therefore included the cortex in the medial bank of the lateral orbital sulcus. The lesion also covered the entire region between the medial and lateral orbital sulci, and extended medially up to the lateral bank of the rostral sulcus. The anterior (rostral) border of the lesion was an imaginary line passing between the anterior tips of the medial and lateral orbital sulci. The posterior limit of the lesions was an imaginary line passing anterior to the posterior tips of the lateral and medial orbital sulci. The intended lesion included the cortex in cytoarchitectonic areas 11, 13 and 14 on the orbital surface[35]. In all three OFC-lesioned monkeys, the lesion covered the intended regions, however in two monkeys there was extremely slight unilateral damage beyond the intended lateral boundary of the lesion and in all three monkeys the lesions did not extent as far medially as intended[25].

**Superior dorso-lateral prefrontal cortex (sdlPFC) lesion.** The extent of intended lesions in the sdlPFC group (Figs. 1b, S7) included the cortex on the most dorsal areas on the lateral prefrontal cortex. The ventral limit of the lesion started 1 mm dorsal to the principal sulcus (Figs. 2b, 5). The lesion extended dorsally up to the longitudinal fissure. Therefore, the lesion included the lateral part of the cytoarchitectonic area 9 and the dorsal portions of areas 46 and 9/46[35]. However, the lesion did not include the cortex within the principal sulcus area and therefore there was no overlap in the lesion extent between the DLPFC and sdlPFC groups. The lesion extent in the sdlPFC group excluded posteriorly located premotor cortex in cytoarchitectonic areas 8A, 8Bd, and 8Bv, and did not extend to the most anterior regions of prefrontal cortex (did not include area 10)[35]. In the three sdlPFC-lesioned monkeys, the lesion covered the intended regions[25].

**Frontopolar cortex lesion.** The extent of lesions in frontopolar cortex was as intended in all animals and covered the dorsal, medial and orbital parts of frontal pole cortex (Figs. 1b, S6)[65]. The posterior limit of the frontopolar cortex lesions was an imaginary line considered at 2 mm posterior to the anterior tip of the principal sulcus. All cortex anterior to this imaginary line was removed.

**Posterior cingulate cortex (PCC) lesion.** The extent of lesions in PCC cortex was as intended in all animals and included cortex on the surface of cingulate gyrus (dorsally limited by the cingulate sulcus) and lower bank of posterior cingulate sulcus (Figs. 1b, S7)[65]. The anterior limit of the PCC lesions was an imaginary vertical line at the most posterior level of the central sulcus and the posterior limit was another imaginary line at the most posterior aspect of the splenium of the corpus callosum, which extended to the posterior end of the cingulate sulcus.

### Histology

At the end of data collection, two animals with DLPFC lesion and four animals with ACC lesions were deeply anaesthetized and then perfused through a cannula in the heart with saline and then by formol-saline solution. Animals' brains were blocked and allowed to sink in sucrose-formalin solution, and subsequently cut in 50 μm sections using a freezing microtome. Every fifth or tenth section was retained and stained with cresyl violet. Histological examination indicated that in all lesion groups the lesion covered the intended cortical regions. The details of lesion extent in each lesion group has been previously reported[25,65].

## Data analyses

Response time (RT) was determined as the interval between the onset of the test items and the first touch of the visual items on the touchscreen (Fig. 1a). For data analyses, all data points (without removal of any outlier), were used for data analyses. In the WCST, we included a response window for initiating and delivering the response for monkeys and therefore all response times falling outside the response window were considered as errors.

Calculation of an index for representing RT variability and comparison between various conditions with different mean RT: The degree of variability might be affected by alterations in the mean RT. Therefore, we calculated Coefficient of RT variation (RT-COV) as the standard deviation of RT divided by the mean RT in each condition[3,4]. We used this method to conclude that the variability significantly increased even when the mean RT (significantly or numerically) increased (which is the case in the OFC-lesioned monkeys) (Figs. S2 and S5), and to conclude that the variability significantly decreased even when the mean RT decreased (which was the case in the ACC-lesioned and DLPFC-lesioned monkeys) (Figs. S2 and S5). We have also reported the results of statistical tests when standard deviation of RT (SD) was used (Figs. S1 and S4). As for the effects of lesion on the difference between the RT variability between cE and cC trials, we emphasized the conclusion only when consistent results were obtained with RT-COV and SD.

For repeated-measure ANOVA test, Mauchly's test of sphericity was applied, and if the sphericity was not met, Greenhouse-Geisser corrections were applied. The RT-COV in testing sessions was used as a data point for ANOVA analyses. In all ANOVA analyses, Partial Eta Squared ($\eta p^2$) indicates the proportion of the variance explained by the effect, and it was reported for each significant effect.

## Reporting summary

Further information on research design is available in the Nature Portfolio Reporting Summary linked to this article.

# Data availability

The original data will be available upon written request to the corresponding author. Source data are provided with this paper.

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

## Acknowledgements
We would like to thank Philip G. F. Browning, Hassan Hoda, Sze C. Kwok, Majid Mahboubi and Adam Phillips for their contributions to the training of monkeys. This work was supported by the Center of Excellence for Integrative Brain Function (Australian Research Council (ARC)) and ARC Discovery project grant; and a grant from the Strategic Research Program for Brain Sciences of the Japan Agency. MJB's contributions were supported by UK MRC grants (G0300817 and MR/W019892/1).

## Author contributions
F.A.M., M.J.B. and K.T. contributed equally to the preparation of manuscript.

## Competing interests
The authors declare no competing interests.
