## [Peer Review File · Nature Communications]

Mapping causal links between prefrontal cortical regions and intra-individual behavioral variability.Reviewer #1 (Remarks to the Author):

Summary: In this manuscript submission, authors report the results of a follow-up analysis of response time data that were collected as part of previously published works by this group exploring the impact of selective aspiration lesions on performance of a computerized version of the Wisconsin Card Sort task. The results reported here seek to enhance the previously published data by providing additional context for interpretation, and also re-analyze new dimensions of previously published results. I commend the authors on the productive re-use of previously collected data! This is an important practice to minimize the use of additional animals for research (part of the 3 Rs). Overall, the manuscript is well composed. However, the reviewer has some concerns that lessen the enthusiasm.

General Comments:

Introduction:

1) The authors situate this manuscript into a very broad context, focusing dominantly on human studies. While helpful to highlight the translational value of this work, it appears to have been done at the expense of omitting an important body of work that more closely relates to the study reported here. Specifically, beyond the previous work by the authors, this manuscript does not include many references, or much discussion/analysis, of other published NHP studies that involve nearly identical (or conceptually similar) tasks. The manuscript should be revised to appropriately acknowledge this body of work, and to better place the current findings into context. Examples of studies that seem to be relevant but are not included:

- a. <https://doi.org/10.1037/a0014723>
- b. <https://doi.org/10.1007/s10071-013-0693-0>
- c. <https://doi.org/10.1523/JNEUROSCI.1766-16.2016>
- d. <https://doi.org/10.3389/fnbeh.2017.00055>
- e. https://doi.org/10.1162/jocn_a_00277
- f. <https://doi.org/10.1523/JNEUROSCI.3690-04.2005>
- g. [https://doi.org/10.1016/S0197-4580\(02\)00054-4](https://doi.org/10.1016/S0197-4580(02)00054-4)
- h. <https://doi.org/10.1037/0735-7044.110.5.872>
- i. <https://doi.org/10.3389/fnsys.2019.00006>
- j. <https://doi.org/10.1016/j.neuroscience.2008.08.047>
- k. <https://doi.org/10.1080/17470210600971485>
- l. <https://doi.org/10.1523/JNEUROSCI.17-23-09285.1997>

Results:

- 1) The statistics are presented in long and hard to follow sentences. It would be helpful if the authors could provide a table summarizing their statistical tests and the results.
- 2) Authors mentioned that corrections were applied in cases that violated assumptions of sphericity, but it does not appear (from the degrees of freedom reported) that any of these were applied. Please clarify.
- 3) The figures provided do not allow for appropriate assessment of the actual lesion extents/locations. Beyond the diagrams showing the intended lesions, please provide more details on the actual lesion extents for each animal in each group involved in this study, as well as an assessment of any unintended damage that occurred outside of the target regions.
- 4) Are there any relationships/correlations between task performance, RT, and lesion extent? For example, do monkeys with larger lesions show bigger effects? Likewise, how do the extents of unintended damage (outside of the intended area) correlate with task performance/RT?
- 5) Figure 1B contains MR images that appear to be identical to those published in Figure 3 of *J Neurosci.* 2014 Aug 13; 34(33): 11016–11031 (case PCC1). Similarly, the diagrams showing lesion extents also appear to be the same as previous publications. I appreciate that there is redundancy in the design (as the same animals were involved in these earlier studies), but it seems inappropriate to duplicate previously published images. Can these images be updated or re-drawn in some way to make a more novel figure?
- 6) Have the number of rule-shift data (Figure 2b,d,f and Figure 3b,d,f) been previously published? These seem to match closely with what was reported in your previous 2009 *Science* publication. Please clarify the differences.
- 7) Referring to Figure 7 and 8. Is there a computational strategy that produced these figures and can be used to predict future behavior? Or are these more illustrative figures of your

interpretations? Please clarify how this was generated.

8) Can Figure 7 and 8 be combined? Likewise, can similar "model" diagrams be produced for all of the lesion groups rather than just a subset? This may improve the reader's ability to understand the utility of the model.

9) In general, for all graphs, my preference is to include brackets to indicate only comparisons that reached the threshold of statistical significance. The figures include many extra brackets and this is a bit confusing/distracting. Of course, please defer to the journal guidelines if this suggestion does not match the submission requirements.

Discussion:

1) In the first paragraph of the discussion, the authors make broad comments regarding decision-making processes reflected by RT. The manuscript could be improved by further refinement/clarification of these ideas. For example, do the authors propose that RT for object recognition tasks (like Delayed Non-Match to Sample) follows the same mechanism? Or working memory tasks (like Delayed Response, or Self-Ordered tasks)?

2) Authors appear to make assumptions that the animals are exclusively using rule-based strategies to respond. Please justify these assumptions and/or provide caveats/alternatives to this interpretation.

3) Please comment on any possible limitations of aspiration lesions, and the possibility that the behavioral effects reported here could be influenced by off-target damage to fibers of passage and/or nearby white matter tracts. This is also relevant for the comments in the discussion relating to resting-state networks.

4) As in the introduction, please revise the discussion to better situate the findings into the broader context of the NHP field and include references/discussion of other NHP studies involving the same/similar task. Please note, I am not asking the authors to necessarily include every study I listed above, just to point out that there is a significant body of research in NHPs that seemingly relates to this manuscript and was overlooked.

Conclusion:

1) The authors appear to construe the specific and localized lesions they made in this study as equivalent to nodes of resting-state fMRI networks, but this conclusion is not supported by the data presented. Are rsfMRI data from these specific animals available to support this claim? Alternatively, if rsfMRI data are not available, authors should provide caveats to this interpretation.

2) Again, here, revisions that situate the current findings into the larger body of NHP work using this (and similar) behavioral paradigms will strengthen the manuscript.

Methods:

1) Please provide more details about the animals involved in this study, as it is quite difficult to follow where the data in this manuscript originated. For example, a table including details about each experimental subject (age, sex, species, lesion group, testing site, and references to other published works with them) would help clarify.

2) Are there any sex or age effects in the data? It is not clear about the demographics of the animals and any impact of age/sex on task performance, nor how these demographic variables compare across groups. Similarly, as there appear to be different species involved (*macaca fuscata* and *macaca mulatta*), and different sites (Oxford and RIKEN), can the authors provide some supplementary analysis (perhaps with the control animals) to demonstrate how performance compares across the two different species/sites?

3) Perhaps I am misunderstanding, but were there sham surgeries performed on the control animals? The authors report only a 2-week recovery post-surgery—this is a short time between surgery and the subsequent testing sessions. Could this be a confound to the behavioral measures? (Since the lesion groups were likely not entirely healed when re-tested?). Please address.

4) Due to slight differences in design, some of the groups appear to have received more opportunities for practice than other groups prior to surgery. The differences in the schedule of testing across groups/animals should be clarified, and also the implications of extra practice discussed.

5) Again, perhaps I am misunderstanding, but since all of the animals were tested before surgery, why not compare everything to baseline? In other words, why are the ACC and DLPFC groups being compared to controls; but the OFC, sdIPFC, Frontopolar, and PCC groups being compared

pre- vs post-lesion?

6) Please provide more details about the cognitive testing. What was the pre-training criteria? Were all animals performing at equivalent levels prior to surgery? Did their performance reach an asymptote pre-surgery? Did all animals receive identical sequences of the task? Were any of the shifts color-to-color or shape-to-shape? Or were they all color-shape/shape-color shifts?

7) Along those lines, are there RT differences between a color->shape shift versus a shape->color shift? (thinking about this reference in particular: Baxter, M. G., & Gaffan, D. (2007). Asymmetry of attentional set in rhesus monkeys learning colour and shape discriminations. *Quarterly Journal of Experimental Psychology*, 60(1), 1–8.)

8) Lines 550: why is mean RT analyzed here instead of RT normalized to SD as in other parts of the manuscript?

9) Please provide further justification for why only cC and cE trials were examined? Explain why you omitted eC and eE?

Minor Comments:

1) Lines 388-392 are copied close to verbatim from lines 78-82.

Reviewer #2 (Remarks to the Author):

This study is a follow-up to a similar report in *Progress in Neurobiology* from the same authors (2022). The current study adds observations from animals with posterior cingulate and frontopolar lesions, adds a deeper temporal analysis (i.e. greater consideration of the outcome of preceding/following trials), and focuses primarily on the coefficient of response time variation – an appropriate metric for the comparisons described. The results are interesting, and may provide insights into changes in response time observed in patient populations after damage to analogous brain regions.

My main concern is that the conclusions are being drawn from such small group sizes, especially for the groups with inter-subject controls. However, I appreciate the value of NHP studies, and understand the impracticality of creating large data sets. It would be appropriate for the authors to note this limitation in the discussion. I'd also like to see the data for each individual monkey to better appreciate the within-group inter-subject variability. Was there a main effect of monkey or interaction between monkey and other factors in any of the ANOVAs? If not, this should be stated.

The authors discuss 'low evidence' vs 'high evidence' conditions (pertaining to trials performed after an error vs those performed after one or more correct trials, respectively). But given that the WCST has 2 rules, should error and correct trials not be equally informative? i.e. an error response provides just as much evidence as a correct response. In fact, if anything, error trials are valuable 'rule switch' cues. Hence this framing of the observation surrounding increased RTs after error trials might benefit from being revisited.

The conclusions drawn are reasonable, but too strong. Other interpretations should be entertained. For example, ACC lesions may not produce impairments in evidence accumulation (especially if the evidence is equal across trials, in which case there'd be no way to test this), but rather cause impairments in behavioural flexibility/set shifting, resulting in perseveration. Increased RTs in the OFC group might be caused by slower evidence accumulation, but they might also be caused by lower motivation due to impaired valuation of the reinforcer.

Ln 407 states that there were 4 ACC lesioned monkeys, but Ln 489 indicates that there were 2.

Ln 413 & 502 indicate 3 OFC monkeys, but line 500 states 4.

Fig. 1b, why are the coronal visualizations of intended PCC lesions depicted differently to those of all other areas? I understand that these have not been verified histologically, but I don't see how that has any bearing on the intended lesion?

Line edits:

Ln 29: 'effects' -> 'effect'

Ln 33: 'for a proper' -> 'to inform a'

Ln 52: 'activities' -> 'activity'
Ln 75: 'occasions' -> 'cases'
Ln 101: it's unusual to introduce figs 7 & 8 before 2 - 6.
Ln 108: 'support' to 'supports'
Ln 148: delete 'while'
Ln 172: 'Fig 3f' -> 'Fig 2f'
Ln 207: '...preceding the current trial, classifications included: eC...'
Ln 327: 'actually found' -> 'observed'
Ln 378: delete 'highly'
Ln 381: '...the causal link between the function of particular brain areas and...'
Ln 444: '...according to the relevant...'
Ln 448: 'corrects' -> 'correct'
Ln 451: " "
Ln 464: '...while monkeys were deeply...'
Ln 468: 'cut and lifted' -> 'opened and reflected'
Ln 474: 'by threads connected to the nearby bones' -> 'by dissolvable sutures connected to the skull'
Ln 507: 'mid-sagittal sulcus' is more correctly referred to as the 'longitudinal fissure'.

Reviewer #3 (Remarks to the Author):

The present paper is a further extension of the impressive collection of papers on prefrontal areas by the same group.

It focuses on the unsolved issue of the direct contribution of prefrontal areas to the variability of reaction times (RT), which is also relevant for pathological states or age-related cognitive decline. They used a large cohort of macaques with restricted bilateral lesions to several areas.

The paper is quite dense but clearly presented. The results clearly contribute to better understand the respective role of several prefrontal areas. Figure 5 is helpful in summing up the results about RT variability.

The results on RT variability are important to the field, however one should note that part of the results (effect of lesions on RT and number of shifts in WCST) are reported in previous papers by the group (and cited in the main text). The paper is remarkable on its own considering the difficulty of carrying lesions in many behaving monkeys. Results clearly show that variability of RT was differentially affected in ACC, PCC and in an opposite way in OFC

Explained by accumulated evidence and decision threshold for rule shifts and a simple model is built. I agree that this model is consistent with the data for both RT variability and RT variability considering correct and error trials.

However, I think that a few clarifications should be made before publication.

1) Age of the animals is obviously crucial and should be reported in the main text (idem for sex which was not included in the design of the study according to the reporting summary).

2) To facilitate comparison with previous studies by the authors, they should indicate whether the monkeys already took part of these former studies (e.g. as described in Kuwabara et al 2014). This is also important to know if part or all of previous results are replicated in a new set of animals (e.g. number of shifts after lesions). For instance, description of lesions (lines 490-492 'the lesion did not extend as far posteriorly as in the other three monkeys to avoid cutting the ascending branches of the anterior cerebral artery') suggest they use the same set of monkeys as in Mansouri et al 2022, macaques from Buckley et al 2009 may be included as well. Note that I am not asking for replication to be compulsory (because it is important to follow the 3R concept of Reduce)

3) ACC lesions and diminution of difference eC- ecC is already mentioned in Kuwabara et al 2014 (their fig 3) and should be cited and discussed (Kuwabara et al did not tested the accumulation of evidence (ecC, eccC)

4) Mansouri et al 2022 show that ACC and OFC lesions impact differentially eCC/eCe and cCC/cCe difference . I think this point should be discussed considering present results on cC trials (cC only

could be cCe or cCc; same for eCe and eCc). Could it be that cC variability is more affected in cCe than cCc ?

5) Frontopolar and PCC controls display much shorter RT and smaller variability than that observed in the other cohort (fig S2 and S4). Do the authors have an explanation for this. For instance, do these monkeys have a more impulsive character trait?

6) I wonder if the results remain stable over the sessions. Maybe some monkeys would need a warming-up in the first part of a daily session for WCST.

7) ACC lesions lead to an increase of 'impulsive' responses. I wonder if impulsive (although supported by the reference cited in introduction) is the only qualitative (monkeys could also become overconfident, this would be in line with a lower threshold in the model). Do the authors note impulsive (faster) responses in control tasks (although not affected by lesions)?

8) Fig 7 is about ACC but should also mention dlPFC, as the lesion in both areas follow the same trend.

9) Line 171-172 One should read 2f instead 'There was also a significant impairment in cognitive flexibility after the OFC lesions (Fig. 3f)25

Point-by-point response to Reviewers' comments

We appreciate the reviewers' constructive and insightful comments.

All changes and additions to the revised manuscript are highlighted in red font.

Reviewer #1:

Summary: In this manuscript submission, authors report the results of a follow-up analysis of response time data that were collected as part of previously published works by this group exploring the impact of selective aspiration lesions on performance of a computerized version of the Wisconsin Card Sort task. The results reported here seek to enhance the previously published data by providing additional context for interpretation, and also re-analyze new dimensions of previously published results. I commend the authors on the productive re-use of previously collected data! This is an important practice to minimize the use of additional animals for research (part of the 3 Rs). Overall, the manuscript is well composed. However, the reviewer has some concerns that lessen the enthusiasm.

Response: We appreciate the reviewer stressing the importance of minimizing the use of non-human primates for research. We would like to mention that assessing the consequence of selective lesions across different prefrontal and frontal regions, within a broader context of decision making and response variability, has further helped to reveal the regional specificity and the underlying mechanisms of previously reported cognitive-behavioural deficits following prefrontal cortical lesions.

General Comments:

Introduction:

1) The authors situate this manuscript into a very broad context, focusing dominantly on human studies. While helpful to highlight the translational value of this work, it appears to have been done at the expense of omitting an important body of work that more closely relates to the study reported here. Specifically, beyond the previous work by the authors, this manuscript does not include many references, or much discussion/analysis, of other published NHP studies that involve nearly identical (or conceptually similar) tasks. The manuscript should be revised to appropriately acknowledge this body of work, and to better place the current findings into context. Examples of studies that seem to be relevant but are not included:

a. <https://doi.org/10.1037/a0014723>

b. <https://doi.org/10.1007/s10071-013-0693-0>

c. <https://doi.org/10.1523/JNEUROSCI.1766-16.2016>

- d. <https://doi.org/10.3389/fnbeh.2017.00055>
 e. https://doi.org/10.1162/jocn_a_00277
 f. <https://doi.org/10.1523/JNEUROSCI.3690-04.2005>
 g. [https://doi.org/10.1016/S0197-4580\(02\)00054-4](https://doi.org/10.1016/S0197-4580(02)00054-4)
 h. <https://doi.org/10.1037/0735-7044.110.5.872>
 i. <https://doi.org/10.3389/fnsys.2019.00006>
 j. <https://doi.org/10.1016/j.neuroscience.2008.08.047>
 k. <https://doi.org/10.1080/17470210600971485>
 l. <https://doi.org/10.1523/JNEUROSCI.17-23-09285.1997>.

Response: We agree with the reviewer's comment that we mainly focused on studies in humans (patients) in assessing the response time (RT) variability. This was mainly because almost all previous studies in monkeys (using different versions of Wisconsin Card Sorting Test (WCST) or set-shifting tasks) did not report the RT variability and mainly focused on alterations in mean RT or overall performance when they studied the neural activity or the effects of various treatments. In the revised manuscript, we cite the relevant studies in non-human primates and further discuss the implications of our findings [Introduction: Page 6-7, Lines 120-126, 129-131, 138-143; Page 20, Conclusion: 424-432, 439-448, Page 21, 451-455].

Results:

1) The statistics are presented in long and hard to follow sentences. It would be helpful if the authors could provide a table summarizing their statistical tests and the results.

Response: In the revised manuscript, we have two new tables (Tables 2 and 3) and moved the details of the ANOVA results from the main text to the tables.

2) Authors mentioned that corrections were applied in cases that violated assumptions of sphericity, but it does not appear (from the degrees of freedom reported) that any of these were applied. Please clarify.

Response: We used IBM SPSS statistical package for statistical analyses. For each repeated-measure ANOVA, Mauchly test of sphericity was conducted and if it was significant, the Greenhouse Geisser correction was considered for reporting the DF and the adjusted significance (which appears on the 'Test of within-subject effects' ANOVA table). For none of the reported comparisons, the Mauchly test was significant and therefore corrections were not necessary.

3) The figures provided do not allow for appropriate assessment of the actual lesion extents/locations. Beyond the diagrams showing the intended lesions, please provide more details

on the actual lesion extents for each animal in each group involved in this study, as well as an assessment of any unintended damage that occurred outside of the target regions.

Response: In the revised manuscript, we have added supplementary figures S6 and S7, which show the actual lesion extent for each animal.

4) Are there any relationships/correlations between task performance, RT, and lesion extent? For example, do monkeys with larger lesions show bigger effects? Likewise, how do the extents of unintended damage (outside of the intended area) correlate with task performance/RT?

Response: The lesion extent was comparable across monkeys in each lesion group because the lesion was visually guided in each animal and all surgeries were performed by the same surgeon. The number of monkeys in each group was limited, which precluded conducting systematic examination of lesion extent and its effects on behavioural measures. However, a qualitative examination of the slight differences in the lesion extent and alterations in behavioural measures did not indicate any correlation.

5) Figure 1B contains MR images that appear to be identical to those published in Figure 3 of J Neurosci. 2014 Aug 13; 34(33): 11016–11031 (case PCC1). Similarly, the diagrams showing lesion extents also appear to be the same as previous publications. I appreciate that there is redundancy in the design (as the same animals were involved in these earlier studies), but it seems inappropriate to duplicate previously published images. Can these images be updated or re-drawn in some way to make a more novel figure?

Response: In the revised manuscript, we have revised the figure to include new diagrams showing lesion extent for PCC and other lesion groups. We also cite, in the Methods [Page 22, Lines 467-470], the previous reports in which details of lesions for each animal group have been described.

6) Have the number of rule-shift data (Figure 2b,d,f and Figure 3b,d,f) been previously published? These seem to match closely with what was reported in your previous 2009 Science publication. Please clarify the differences.

Response: In the original manuscript, we presented the rule-shift data alongside RT variability for each group, while citing the related previously published reports, to show whether and how alterations in RT variability were accompanied by deficits in cognitive flexibility. In the revised manuscript, we have removed these figures and report the deficit in cognitive flexibility (and cite the relevant report) only in the main text [Results: Page 8-10, lines 164, 183, 193-194, 201, 209, 219; Methods: Page 22, Lines 467-470].

7) Referring to Figure 7 and 8. Is there a computational strategy that produced these figures and can be used to predict future behavior? Or are these more illustrative figures of your interpretations? Please clarify how this was generated.

Response: Figure 7 and 8 mainly illustrate our proposed changes in ‘decision threshold’ and ‘the rate of evidence accumulation’ in the context of decision-making processes. The proposed scheme predicts the findings in this manuscript and also conforms well with previously observed behavioural alterations following lesions within ACC or OFC. In the supplementary material of the revised manuscript we have further elaborated the computational background.

8) Can Figure 7 and 8 be combined? Likewise, can similar “model” diagrams be produced for all of the lesion groups rather than just a subset? This may improve the reader’s ability to understand the utility of the model.

Response: In the revised manuscript, Figures 7 and 8 are combined and presented as Figure 7.

9) In general, for all graphs, my preference is to include brackets to indicate only comparisons that reached the threshold of statistical significance. The figures include many extra brackets and this is a bit confusing/distracting. Of course, please defer to the journal guidelines if this suggestion does not match the submission requirements.

Response: We appreciate Reviewer’s comment on only showing the significant p values. However, we assumed reporting significant and also non-significant values would be informative and ease comparisons between groups. It is also recommended by Nature publishing group journals.

Discussion:

1) In the first paragraph of the discussion, the authors make broad comments regarding decision-making processes reflected by RT. The manuscript could be improved by further refinement/clarification of these ideas. For example, do the authors propose that RT for object recognition tasks (like Delayed Non-Match to Sample) follows the same mechanism? Or working memory tasks (like Delayed Response, or Self-Ordered tasks)?

Response: Our current view is that when the sensory-motor aspects of the task remains the same, the RT would mainly be influenced by the two main aspects of decision process, namely the evidence accumulation and the decision threshold. In a changing and complex environment, such as the WCST, where accumulation of evidence for available choices requires executive control (i. e. allocation of cognitive resources and integration of multiple sources of information, such as the working memory of the currently relevant rule and the decision outcome), the fluctuation of executive control will affect the rate of evidence accumulation and consequently correlate with RT

fluctuations. In addition, alterations in decision threshold will also affect the RT and its fluctuation. Therefore, for complex and demanding tasks such as the WCST, RT would reflect these aspects of decision process, however for simple and very well-practiced tasks this association might be weaker. Even for DNMS or self-ordered tasks, whenever the task is difficult due to factors like a long delay or requirement of fine discrimination, RT would fluctuate. In the revised manuscript, we have added an additional section to further discuss the link between RT and executive control fluctuations [Page 14, Discussion: Lines 288-292].

2) Authors appear to make assumptions that the animals are exclusively using rule-based strategies to respond. Please justify these assumptions and/or provide caveats/alternatives to this interpretation.

Response: In the WCST analog used in our studies, the sample, the test items and their position were randomly changed trial-by-trial. Furthermore, there was no cue to the relevant rule or its frequent changes. Therefore, monkeys had to find the relevant rule by trial and error and keep it in working memory to enable applying the rule in consecutive trials and attain the shift criterion. We also used combination of 6 colours and 6 shapes to make a set of 36 items. Considering this task design and the fact that control monkeys could shift between rules (attain high performance following each rule shift) by committing a limited number of errors, it is very unlikely that monkeys implemented an association-based strategy to adapt to frequent rule shifts. In addition, in earlier studies we showed that monkeys could generalize the rule to novel items, which indicated they were applying the rules, but not association-based strategy, for action selection in the WCST [Mansouri et al. 2002]. In the revised manuscript, we have added this point to the Introduction and cited the relevant publications [Page 6, Introduction: Lines 115-119].

3) Please comment on any possible limitations of aspiration lesions, and the possibility that the behavioral effects reported here could be influenced by off-target damage to fibers of passage and/or nearby white matter tracts. This is also relevant for the comments in the discussion relating to resting-state networks.

Response: This is an important point regarding lesion-inactivation studies in non-human primates. The visually-guided aspiration lesion technique is still one of the most suitable and currently-available procedure in macaque monkeys to address the goal of this study. Inactivation studies using chemicals such as GABA agonists (e.g. Muscimol) or neurotoxic compounds (e.g. Ibotenic acid) may allow localized inactivation or death of neurons at the injection sites, but cannot mimic the effects of complete disruption of large cortical regions, which were targeted in this study. Increasing the concentration or volume of injections for making complete lesions would cause

certain involvement of nearby cortical areas, which would prevent proper interpretation of the lesion effects on cognitive functions. In addition, new molecular-genetic techniques for controlled inactivation/activation of neuronal populations (e.g. DREADD or optogenetics), which have been used in rodent models are still being developed for primates. The success rate in transferring the genetic codes to primate neurons is still low and therefore not feasible for complete and bilateral inactivation of deep and large cortical areas, which were targeted in this study.

The possibility of damage to the immediately underlying white matter, during aspiration of cortex, cannot be ruled out. However, we made utmost care during surgery (visually guided through microscope) to avoid any deep damage to the underlying white matter. Histological examination of lesion extent did not show any deep or significant involvement of underlying white matter. Therefore, the possibility of a significant damage to major fascicles was very low. In the Method section of the revised manuscript, we have highlighted these limitations in lesion/inactivation studies in non-human primates [Page 26, Methods: Lines 562-576].

4) As in the introduction, please revise the discussion to better situate the findings into the broader context of the NHP field and include references/discussion of other NHP studies involving the same/similar task. Please note, I am not asking the authors to necessarily include every study I listed above, just to point out that there is a significant body of research in NHPs that seemingly relates to this manuscript and was overlooked.

Response: In the revised manuscript, we cite the relevant studies in non-human primate studies and further discuss the implications of our findings in relation to those studies using various analogues of the WCST and set-shifting tasks [Page 6-7, Introduction: Lines 120-126, 129-131, 138-143; Pages 20-21, Conclusion: 424-432, 439-448, 451-455].

Conclusion:

1) The authors appear to construe the specific and localized lesions they made in this study as equivalent to nodes of resting-state fMRI networks, but this conclusion is not supported by the data presented. Are rsfMRI data from these specific animals available to support this claim? Alternatively, if rsfMRI data are not available, authors should provide caveats to this interpretation.

Response: We agree that the functional correspondence in default mode network, between humans and non-human primates, has not been fully established yet. Our previous description implied that we considered the same neural nodes as functional homologues of default mode network in humans. In the revised manuscript [Page 6, Introduction: Lines 131-137; Page 19, Discussion: 400-403], we have revised the related sentences and clarified these points while citing the highly relevant studies

in non-human primates.: *“In humans, posterior cingulate cortex (PCC) and frontopolar cortex comprise the main nodes of the default mode network. Resting state functional connectivity studies have suggested the presence of a comparable network in non-human primates, although the functional homology remains to be established.”*

2) Again, here, revisions that situate the current findings into the larger body of NHP work using this (and similar) behavioral paradigms will strengthen the manuscript.

Response: In the revised manuscript, we cite the relevant studies in non-human studies and further discuss the implications of our findings for interpreting the relevant studies, which have examined the role of prefrontal cortical regions in cognitive flexibility [Page 6, Introduction: Lines 120-126, 129-131, 138-143; Page 20, Conclusion: 424-432, 439-448, 451-455].

Methods:

1) Please provide more details about the animals involved in this study, as it is quite difficult to follow where the data in this manuscript originated. For example, a table including details about each experimental subject (age, sex, species, lesion group, testing site, and references to other published works with them) would help clarify.

Response: In the revised manuscript, we have included Table 1, which includes the detailed demographic information regarding the involved monkeys [Page 22, Methods: Lines 467-470].

2) Are there any sex or age effects in the data? It is not clear about the demographics of the animals and any impact of age/sex on task performance, nor how these demographic variables compare across groups. Similarly, as there appear to be different species involved (macaca fuscata and macaca mulatta), and different sites (Oxford and RIKEN), can the authors provide some supplementary analysis (perhaps with the control animals) to demonstrate how performance compares across the two different species/sites?

Response: In the revised manuscript, we have included Table 1, which includes the detailed demographic information (sex, species) and monkeys' training and testing locations [Page 22, Methods: Lines 467-470]. All 21 macaque monkeys were young adults, however the exact age of some of these monkeys were not available and therefore age-based comparisons were not feasible. There was only 1 female monkey (in DLPFC group) and therefore sex dependency of effects could not be assessed. Equal numbers of macaque fuscata and mulatta were assigned to control, DLPFC and ACC groups. The OFC group included two mulatta and one fuscata. The sdIPFC group included one mulatta and two fuscata. The same software and training schedule/testing were used

for all monkeys and the same surgeon performed all lesion surgeries in both institutions (Oxford University and RIKEN institute).

3) Perhaps I am misunderstanding, but were there sham surgeries performed on the control animals? The authors report only a 2-week recovery post-surgery—this is a short time between surgery and the subsequent testing sessions. Could this be a confound to the behavioral measures? (Since the lesion groups were likely not entirely healed when re-tested?). Please address.

Response: No sham surgery was performed on Control monkeys, considering that these monkeys were later assigned to two separate lesion groups (OFC and sdIPFC). All monkeys in the lesion groups had a post-operative rest period for about 2 weeks and then post-lesion data collection was commenced. All control animals had an equal duration of rest between the pre-operative and post-operative testing. All animals in control and lesion groups performed post-operative ‘Control tasks’ in which the relevant rule remained the same in the entire daily testing session (no rule shift was imposed). All monkeys performed the control tasks at very high level (more than 85% correct) indicating that they had no perceptual-motor deficits [Page 25, Methods: Lines 540-545]. The effects of lesion appeared when monkeys performed the WCST (rule shift was required). Also, the monkeys with lesions in Frontopolar or PCC did not show changes in the number of rule shifts per day. Therefore, it is highly unlikely that monkeys’ altered behaviour resulted from non-specific factors (such as surgery or anaesthesia).

4) Due to slight differences in design, some of the groups appear to have received more opportunities for practice than other groups prior to surgery. The differences in the schedule of testing across groups/animals should be clarified, and also the implications of extra practice discussed.

Response: After completion of the post-operative testing with the Control monkeys in the first cohort, these 6 monkeys were assigned to OFC and sdIPFC groups (3 monkeys in each group). Therefore, monkeys in the OFC and sdIPFC groups had performed more sessions compared to the ACC and DLPFC groups at the time of operation. However, the effects of lesions within the OFC or in the sdIPFC on behavioural measures were assessed by comparing the pre-lesion and post-lesion performance (repeated-measure design). The pre-lesion data were collected just before the lesion surgeries. Therefore, the additional practice (while they served as the Control group) could not explain the effects of lesions in the OFC or sdIPFC. In fact, the lesions in OFC had the most severe adverse effects on cognitive flexibility of monkeys (decreased the number of post-operative rule-shifts) [Buckley et al. 2009].

In the second cohort of 7 monkeys, the lesions were first made in frontopolar cortex in 4 monkeys while the other 3 monkeys remained unoperated. The 3 unoperated monkeys also performed the WCST while the 15 pre- and 15 post-operative testing sessions were completed for the frontopolar-lesioned monkeys. After completion of the post-operative sessions in frontopolar-lesioned monkeys, the pre-lesion data were collected in the 3 unoperated animals and then they received lesions in the PCC. Therefore, the 3 monkeys in the PCC group had more practice with the WCST (compared to the frontopolar-lesioned group). However, as mentioned above, the effects of lesions within the frontopolar cortex or in the PCC were assessed by comparing the pre-lesion and post-lesion performance (repeated-measure design). Therefore, the additional practice could not explain the effects of lesions in the PCC-lesioned monkeys. In the revised manuscript, we have added these explanations to the Methods sections [Page 23, Methods: Lines 491-496, 504-511].

5) Again, perhaps I am misunderstanding, but since all of the animals were tested before surgery, why not compare everything to baseline? In other words, why are the ACC and DLPFC groups being compared to controls; but the OFC, sdIPFC, Frontopolar, and PCC groups being compared pre- vs post-lesion?

Response: We appreciate reviewer's comment and agree that pre-lesion vs post-lesion comparisons across all lesion groups would have been informative. Our analyses are restricted by a limitation in our studies with the first cohort, which had the first stage (for ACC and DLPFC lesions) and second stage (for OFC and sdIPFC lesions). In the pre-lesion sessions of the first stage, we did not register the monkeys' trial-by-trial RT (the data collection software only registered the mean RT and number of rule-shifts). Allocation of monkeys to different groups was done based on the mean number of rule-shifts. In the post-lesion sessions of the first stage and in both pre- and post-lesion sessions of the second stage, the monkeys' trial-by-trial RT was recorded. In studies with the second cohort (for frontopolar and PCC lesions), the trial-by-trial RT was recorded in both pre-lesion and post-lesion sessions.

6) Please provide more details about the cognitive testing. What was the pre-training criteria? Were all animals performing at equivalent levels prior to surgery? Did their performance reach an asymptote pre-surgery? Did all animals receive identical sequences of the task? Were any of the shifts color-to-color or shape-to-shape? Or were they all color-shape/shape-color shifts?

Response: All animals at RIKEN institute and Oxford University were trained by the same software following the same training schedule. Animals were trained in 10 stages; The details of the training schedule and related criteria have been described in the supplementary material of Buckley et al. (Science Vol 325, Issue 5936 (2009)). In the later stages of the training, all animals achieved a

criterion (85% correct) for matching with colour or with shape rules within a daily session. The relevant rule alternated between colour and shape across days. Then, monkeys moved to the next stage in which the rule was alternated only once within the daily session and they had to achieve a criterion (90% correct in 40 trials). The starting rule of each session changed across days. After completion of this stage, they moved to the next stage of training with a version of the WCST in which the relevant rule was changed frequently and they had to achieve 85% correct in 20 consecutive trials. All animals achieved this criterion before pre-lesion data collection. The performance of animals differed in the final, and most challenging, version of the WCST. Each animal had at least 3 rule shifts within a daily testing session and then pre-lesion data were collected in 15 pre-lesion sessions and 15 post-lesion sessions. According to the pre-lesion performance of monkeys, they were assigned to Control or lesion groups so that the range and mean of the numbers of pre-lesion shifts between rules were comparable between groups. In the revised manuscript, we have included Table 1 to describe the demographic information and training history of each animal and also cited the relevant studies [Page 22, Methods: Lines 467-470]. All animals were also tested both pre- and post-operatively on two control tasks that were designed to assess whether their basic perceptual, motor, attentional, and stimulus matching skills were intact [Page 25, Methods: Lines 539-545].

7) Along those lines, are there RT differences between a color->shape shift versus a shape->color shift? (thinking about this reference in particular: Baxter, M. G., & Gaffan, D. (2007). Asymmetry of attentional set in rhesus monkeys learning colour and shape discriminations. Quarterly Journal of Experimental Psychology, 60(1), 1–8.).

Response: We appreciate the reviewer's comment and raising this important point. In fact, monkeys and humans show significant behavioural advantage (dimensional bias) with one of the rules (colour or shape) when they perform similar computerized analogues of the WCST. Monkeys and humans show significant bias to shape and colour rule, respectively. Lesions in monkeys' prefrontal cortical regions did not obliterate these dimensional biases. The details of these dimensional biases in trichromatic primate species have been reported in our recent paper [Mansouri et al. Cerebral Cortex 30, 85–99 (2020)]. We have also found these dimensional biases when monkeys and humans perform stop-signal task (bias in inhibition ability) [Ghasemian et al. Animal Cognition 24, 815-828 (2021)] and working memory tasks [Fehring et al. Scientific Reports 12, 5335 (2022)]. In our ongoing studies we further examine the neural substrate and underlying neural mechanisms of such biases.

8) Lines 550: why is mean RT analyzed here instead of RT normalized to SD as in other parts of the manuscript?

Response: We apologize for this oversight (typo) and have corrected that to RT-COV (Coefficient of RT variation) in the revised manuscript.

9) Please provide further justification for why only cC and cE trials were examined? Explain why you omitted eC and eE?

Response: The number of eE trial sequences was very limited (particularly in the Control monkeys and in pre-lesion sessions) and therefore calculation of behavioural variability in these trials was not much feasible or reliable. The monkeys' behaviour in eC trials had particular characteristics, which made them conceptually different from cC and cE trials. Monkeys' performance after an error trial dropped to 50% correct, which for a highly trained monkey means chance level (selecting between colour or shape rule). This implies that there was a resetting of performance after committing a perseverative error and therefore performance in the next correct trial could not be easily associated with a correct rule selection. Therefore, we limited the comparisons to cC and cE trials in this manuscript. We have added additional explanations for trial selection in the revised manuscript [Pages 6-7, Introduction: Lines 138-143].

Minor Comments:

1) Lines 388-392 are copied close to verbatim from lines 78-82.

Response: We apologize for this repetition (it is removed from the Method section of the revised manuscript).

Reviewer #2 (Remarks to the Author):

This study is a follow-up to a similar report in Progress in Neurobiology from the same authors (2022). The current study adds observations from animals with posterior cingulate and frontopolar lesions, adds a deeper temporal analysis (i.e. greater consideration of the outcome of preceding/following trials), and focuses primarily on the coefficient of response time variation – an appropriate metric for the comparisons described. The results are interesting, and may provide insights into changes in response time observed in patient populations after damage to analogous brain regions.

My main concern is that the conclusions are being drawn from such small group sizes, especially for the groups with inter-subject controls. However, I appreciate the value of NHP studies, and understand the impracticality of creating large data sets. It would be appropriate for the authors to note this limitation in the discussion. I'd also like to see the data for each individual monkey to better appreciate the within-group inter-subject variability. Was there a main effect of monkey or interaction between monkey and other factors in any of the ANOVAs? If not, this should be stated.

Response: We appreciate Reviewer's positive view toward this study in non-human primates. As the Reviewer mentioned, compared to studies in rodents, usually limited numbers of monkeys are used in studies with non-human primate models. We have acknowledged this limitation in the Conclusion section of the revised manuscript.

We would like to mention that, as a norm in studies with non-human primates, we collected multiple sessions in each animal before (15 daily sessions) and after (15 daily sessions) selective brain lesions. In the revised manuscript (Figure 2b, 2d, 2f, 3b, 3d and 3f), we have included the values in each session for each monkey. These new figures show the RT-COV of individual monkeys in 15 pre-lesion (red color) and 15 post-lesion (black color) sessions for Control and lesion groups, respectively. The values for each monkey appear with a distinct marker shape.

Our analyses were restricted by a limitation in our studies with the first cohort, which had the first stage (for ACC and DLPFC lesions) and second stage (for OFC and sdLPFC lesions). In the pre-lesion sessions of the first stage, we did not register the monkeys' trial-by-trial RT (the data collection software only registered the mean RT and number of rule-shifts). Allocation of monkeys to different groups was done based on the mean number of rule-shifts. In the post-lesion sessions of the first stage and in both pre- and post-lesion sessions of the second stage, the monkeys' trial-by-trial RT was recorded. Therefore, the performance of ACC-lesioned and DLPFC-lesioned monkeys were compared with Control group monkeys in the first stage, however for OFC-lesioned and sdLPFC-lesioned monkeys (second stage), comparisons were done between the pre-lesion and post-lesion performance (repeated-measure analyses). For comparison of Control and ACC-lesioned (or

DLPFC-lesioned) monkeys, we used a nested ANOVA (Lesion-group x Monkey) in which Monkey was nested within the lesion-group factor. In this nested ANOVA, the interaction effect cannot be examined. The Monkey factor was significant when we compared ACC-lesioned or DLPFC-lesioned monkeys with Control group. This mainly resulted from differences between monkeys, while the difference between Control and ACC-lesioned monkeys (Main effect of Lesion-group) remained highly significant ($p < 0.001$). The difference between DLPFC-lesioned and Control group was marginally significant ($p = 0.041$) and therefore we have mainly emphasized the significant alterations in ACC-lesioned monkeys in our conclusions and the related models.

For all pre-lesion vs post-lesion comparisons of RT variability (RT-COV) in the OFC, sdlPFC, PCC, and frontopolar groups, the Lesion-group x Monkey interaction was not significant. The explanation and the details of ANOVA results in each group have been revised and included in new Tables 2 and 3 in the revised manuscript.

The authors discuss 'low evidence' vs 'high evidence' conditions (pertaining to trials performed after an error vs those performed after one or more correct trials, respectively). But given that the WCST has 2 rules, should error and correct trials not be equally informative? i.e. an error response provides just as much evidence as a correct response. In fact, if anything, error trials are valuable 'rule switch' cues. Hence this framing of the observation surrounding increased RTs after error trials might benefit from being revisited.

Response: We found that in monkeys performing the WCST, after committing each error, monkeys' performance dropped to around 50% [Figure 3D in Buckley et al. Science 325 (2009)]. For highly trained monkeys, this is the chance level (selecting between colour or shape rule) because monkeys rarely had non-perseverative errors (selecting the item, which did not match the sample in colour or shape). Therefore, there was a kind of re-setting after committing a perseverative error and then monkeys further improved their performance with the new rule after each correct (rewarded) trial [Figures 3D and 4A in Buckley et al. Science 325 (2009)]. These imply that monkeys actually learned from their errors that the rule had been changed, however they did not immediately adapt the new rule (different from humans performing the WCST: Mansouri et al. Cerebral Cortex 2020). Instead, they further learned from consecutive correct trials (accumulated further evidence) and improved their performance until they achieved the shift criterion. Therefore, in Figure 6 of the current manuscript we examined RT in the trial sequences following the commission of error (eC, ecC and eccC), which indicates gradual performance improvement after consecutive correct (rewarded) trials. We have added additional explanations in the revised manuscript [Page 15, Discussion: Lines 311-316].

The conclusions drawn are reasonable, but too strong. Other interpretations should be entertained. For example, ACC lesions may not produce impairments in evidence accumulation (especially if the evidence is equal across trials, in which case there'd be no way to test this), but rather cause impairments in behavioural flexibility/set shifting, resulting in perseveration. Increased RTs in the OFC group might be caused by slower evidence accumulation, but they might also be caused by lower motivation due to impaired valuation of the reinforcer.

Response: We appreciate the reviewer's insight on this important point. We have proposed that the decrease in decision threshold in ACC-lesioned monkeys is the most parsimonious explanation of our findings, because it consistently explains the decrease in RT variability and other aspects of their behaviour in the WCST; and also conforms well with previous findings in humans. As raised by the Reviewer, ACC-lesioned monkeys might also have deficits in other aspects of executive control such as assessment of action outcome. In the Discussion/Conclusion section of the revised manuscript, we have further discussed these alternative interpretations [Page 20, Lines 439-448].

For OFC-lesioned monkeys, the deficits in evidence accumulation can be inferred from different aspects of their performance in the WCST, which also explains their heightened behavioural variability. OFC-lesioned monkeys did not show any significant deficit in Control tasks in which the rule remained the same in the entire daily session. This indicates that their basic perceptual, motor, motivational, valuation of reward and even ability to apply individual matching rules remained intact. However, as we have discussed in the manuscript [Page 20, Lines 439-441; Page 18, Lines 381-392], in the WCST, OFC-lesioned monkeys showed deficits in improving their performance after receiving a single reward, which suggests that they were impaired in assessing their behavioural outcome, and consequently in evidence accumulation, for improving their rule-based behaviour.

Ln 407 states that there were 4 ACC lesioned monkeys, but Ln 489 indicates that there were 2.

Response: Here, we meant in two out of the four ACC-lesioned monkeys, the lesions were complete and as intended. We have revised the sentence to clarify this point.

Ln 413 & 502 indicate 3 OFC monkeys, but line 500 states 4.

Response: We apologize for this oversight (typo) and corrected that in the revised manuscript.

Fig. 1b, why are the coronal visualizations of intended PCC lesions depicted differently to those of all other areas? I understand that these have not been verified histologically, but I don't see how that has any bearing on the intended lesion?

Response: In PCC-lesioned group, the lesion extent was verified by structural MRI. In the revised manuscript, we have replaced the images with corresponding diagrams for depicting the intended and actual lesions for all lesion groups.

Line edits:

Ln 29: 'effects' -> 'effect'

Ln 33: 'for a proper' -> 'to inform a'

Ln 52: 'activities' -> 'activity'

Ln 75: 'occasions' -> 'cases'

Ln 101: it's unusual to introduce figs 7 & 8 before 2 – 6.

Ln 108: 'support' to 'supports'

Ln 148: delete 'while'

Ln 172: 'Fig 3f' -> 'Fig 2f'

Ln 207: '...preceding the current trial, classifications included: eC...'

Ln 327: 'actually found' -> 'observed'

Ln 378: delete 'highly'

Ln 381: '...the causal link between the function of particular brain areas and...'

Ln 444: '...according to the relevant...'

Ln 448: 'corrects' -> 'correct'

Ln 451: “ “

Ln 464: '...while monkeys were deeply...'

Ln 468: 'cut and lifted' -> 'opened and reflected'

Ln 474: 'by threads connected to the nearby bones' -> 'by dissolvable sutures connected to the skull'

Ln 507: 'mid-sagittal sulcus' is more correctly referred to as the 'longitudinal fissure'.

Response: We appreciate the reviewer's suggestions and have applied all of these changes in the revised manuscript.

Reviewer #3 (Remarks to the Author):

The present paper is a further extension of the impressive collection of papers on prefrontal areas by the same group. It focuses on the unsolved issue of the direct contribution of prefrontal areas to the variability of reaction times (RT), which is also relevant for pathological states or age-related cognitive decline.

They used a large cohort of macaques with restricted bilateral lesions to several areas.

The paper is quite dense but clearly presented. The results clearly contribute to better understand the respective role of several prefrontal areas. Figure 5 is helpful in summing up the results about RT variability.

The results on RT variability are important to the field, however one should note that part of the results (effect of lesions on RT and number of shifts in WCST) are reported in previous papers by the group (and cited in the main text).

The paper is remarkable on its own considering the difficulty of carrying lesions in many behaving monkeys. Results clearly show that variability of RT was differentially affected in ACC, PCC and in an opposite way in OFC.

Explained by accumulated evidence and decision threshold for rule shifts and a simple model is built. I agree that this model is consistent with the data for both RT variability and RT variability considering correct and error trials.

However, I think that a few clarifications should be made before publication.

1) Age of the animals is obviously crucial and should be reported in the main text (idem for sex which was not included in the design of the study according to the reporting summary).

Response: In the revised manuscript, we have included Table 1, which describes the detailed demographic information (sex, species) and monkeys' training and testing locations [Page 22, Methods: Lines 467-470]. All 21 macaque monkeys were young adults, however the exact age of some of these monkeys were not available and therefore age-based comparisons were not feasible. There was only 1 female monkey (in DLPFC group) and therefore sex dependency of effects could not be assessed. Equal numbers of macaque fuscata and mulatta were assigned to control, DLPFC and ACC groups. The OFC group included two mulatta and one fuscata. The sdIPFC group included one mulatta and two fuscata species.

2) To facilitate comparison with previous studies by the authors, they should indicate whether the monkeys already took part of these former studies (e.g. as described in Kuwabara et al 2014). This is also important to know if part or all of previous results are replicated in a new set of animals (e.g. number of shifts after lesions). For instance, description of lesions (lines 490-492 'the lesion

did not extend as far posteriorly as in the other three monkeys to avoid cutting the ascending branches of the anterior cerebral artery') suggest they use the same set of monkeys as in Mansouri et al 2022, macaques from Buckley et al 2009 may be included as well. Note that I am not asking for replication to be compulsory (because it is important to follow the 3R concept of Reduce).

Response: In the revised manuscript, we have added Table 1 and further explanation of the history of animals in this study and cited the previous relevant studies [Page 22, Methods: Lines 467-470]. We have also added supplementary figures S6 and S7, which include detailed information regarding the lesion extent in each animal.

3) ACC lesions and diminution of difference eC- ecC is already mentioned in Kuwabara et al 2014 (their fig 3) and should be cited and discussed (Kuwabara et al did not tested the accumulation of evidence (ecC, eccC)

Response: The lower decision threshold also explains some of our previous findings in ACC-lesioned monkeys. In the revised manuscript, we cite Kuwabara et al. (2014) and further discuss the implications of our proposed model for interpreting the previously reported behaviour of ACC-lesioned monkeys [Pages 16-17, Discussion: Lines 345-352; Page 20, Conclusion: Lines 441-448].

4) Mansouri et al 2022 show that ACC and OFC lesions impact differentially eCC/eCe and cCc/cCe difference . I think this point should be discussed considering present results on cC trials (cC only could be cCe or cCc; same for eCe and eCc). Could it be that cC variability is more affected in cCe than cCc ?

Response: We appreciate the reviewer for raising this issue. We examined whether RT-COV (mean RT/SD) differed between cCc and cCe sequences in Control monkeys and found that the main effect of Trial-sequence (cCc/cCe) was significant [$F(1,84) = 5.77, p = 0.019$]: the RT-COV was higher in cCe trials. The higher RT variability in cCe trials suggest that the executive control was at lower levels in these trials, and in fact the monkey committed an error in the following trial. However, when we examined the effects of lesions in ACC, DLPFC, sdlPFC, Frontopolar or PCC, there was no significant interaction between Lesion and Trial-sequence factors, which suggests that lesions in these brain regions did not affect the difference in RT variability between cCc and cCe sequences. In the revised manuscript, we have added a new section to the Results to describe the difference in RT variability between cCc and cCe trials [Page 12-13, Results: Lines 255-271].

5) Frontopolar and PCC controls display much shorter RT and smaller variability than that observed in the other cohort (fig S2 and S4). Do the authors have an explanation for this. For instance, do these monkeys have a more impulsive character trait?

Response: As the Reviewer mentioned, monkeys in the second cohort had shorter mean RT compared to the monkeys in the first cohort (Fig. S2). Whereas the RT-COV in the pre-lesion test of the Frontopolar group was comparable with those of the first cohort, the RT-COV in the pre-lesion test of the PCC group was smaller than those of the first cohort (Figs. 2 and 3). As can be seen in Figs. 2b, 2d, 2f, 3b, 3d, and 3f for the variance in RT-COV, the mean RT and RT-COV largely varies among individual monkeys. We assume that the differences between the monkeys in the first and second cohort were due to individual differences.

6) I wonder if the results remain stable over the sessions. Maybe some monkeys would need a warming-up in the first part of a daily session for WCST.

Response: The monkeys were highly trained (at least 1 year) when pre-lesion data collection was initiated. All the monkeys completed 300 trials per daily session in all the days of pre-lesion and post-lesion tests. Overall, the monkeys were more motivated to perform the test at the beginning of the daily session because they were under food scheduling and received a banana-flavoured food pellet for each correct response. They also continued to perform well until the end of the daily session, probably because a lunchbox (containing their daily food) was placed just next to the touchscreen and automatically opened when 300 trials were completed. We did not notice any systematic change in behaviour during the daily session, which suggested the necessity for warming up or appearance of fatigue toward the end of the session.

7) ACC lesions lead to an increase of ‘impulsive’ responses. I wonder if impulsive (although supported by the reference cited in introduction) is the only qualificative (monkeys could also become overconfident, this would be in line with a lower threshold in the model). Do the authors note impulsive (faster) responses in control tasks (although not affected by lesions)?

Response: In the Control tasks in which the rule did not change within the daily session, the response time of ACC-lesioned monkeys was unchanged [Page 25, Methods: Lines 539-545]. They showed faster RT only in the WCST (which included frequent rule shifts) [Buckley et al. Science 325 (2009)]. Therefore, the impulsive responses of ACC-lesioned monkeys in the WCST cannot be simply explained by a general increase in their motivation or confidence.

8) Fig 7 is about ACC but should also mention dlPFC, as the lesion in both areas follow the same trend.

Response: We have included this explanation to Figure 7 (in the revised manuscript it is a combination of previous Figures 7 and 8). In our conclusions regarding the decrease in RT-COV and its underlying mechanisms, we have mainly focused on ACC-lesioned monkeys because the

effects of ACC lesion were highly significant, but it was only marginally significant in the DLPFC-lesioned monkeys. This explanation has also been added to the Discussion section of the revised manuscript [Pages 16-17, Discussion: Lines 345-347].

9) Line 171-172 One should read 2f instead 'There was also a significant impairment in cognitive flexibility after the OFC lesions (Fig. 3f)25

Response: We have corrected this typo in the revised manuscript.

Reviewer #1 (Remarks to the Author):

The authors have fully addressed my earlier comments. The manuscript is greatly improved and I have no further comments at this time.

Reviewer #2 (Remarks to the Author):

The authors have made extensive revisions to the manuscript, created new figure content, and provided tables to simplify comparisons. All of my concerns have been addressed.

Reviewer #3 (Remarks to the Author):

The answers to the questions I have raised are completely satisfactory. The authors have provided convincing explanations, in particular by means of detailed tables and modified figures.

I have also read the answers to reviewers 1 and 2 and I think the authors have answered correctly and in detail.